

# A comparison of supraglacial meltwater features throughout contrasting melt seasons: Southwest Greenland

Emily Glen[1], Amber Leeson[1], Alison F. Banwell[3], Jennifer Maddalena[1], Diarmuid Corr[1], Brice Noël[4], Malcolm McMillan[1, 2]

[1] Lancaster Environment Centre, Lancaster University, Lancaster, UK
[2] UK Centre for Polar Observation & Modelling, Centre for Excellence in Environmental Data Science, Lancaster University, Lancaster, UK
[3] Cooperative Institute for Research in Environmental Sciences (CIRES), University of Colorado Boulder, Boulder, USA
[4] Laboratoire de Climatologie et Topoclimatologie, University of Liège, Liège, Belgium

*Correspondence to*: Emily Glen (e.glen@lancaster.ac.uk)

**Abstract**. The Greenland Ice Sheet is losing mass through increased melting and solid ice discharge. Supraglacial meltwater
features (e.g., lakes, rivers and slush) are becoming more abundant as a result of the former and are implicated as a control on the latter when they drain. It is not yet clear, however, how this system will respond to future climate changes, and it is likely that melting will continue to increase as the Arctic continues to warm. Here, we use Sentinel-2 and Landsat 8 satellite imagery to compare meltwater features in the Russell/Leverett glacier catchment in a high (2019) and a comparatively low (2018) melt year. We find that in the higher melt year: 1) surface meltwater features form and drain, at ~200 and ~400 m higher elevations, 20 respectively, 2) that small lakes (< 0.0495 km²) - typically disregarded in previous studies - are more prevalent and 3) that slush is more widespread. This is important because we show that all three of these sets of features are associated with transient increases in velocity when they drain, and because refreezing of slush can create ice slabs, which inhibit the storage of meltwater in the porous firn and promote surface ponding and runoff in future years. Interestingly, we also identify the potential occurrence of a cascading lake drainage event in the higher melt year, which also appears to perturb ice velocity. Our study 25 therefore suggests that previously poorly mapped and under-studied features (such as small lakes and slush) are actually important in terms of their impact on ice flow and supraglacial runoff, and thus on global sea level rise, in future, warmer, years.

## 1. Introduction

Over recent decades, the Greenland Ice Sheet (GrIS) has undergone substantial mass loss, totalling 4,892 ± 457 Gt of ice from 30 1992 to 2020 (Otosaka et al., 2023). This has arisen through a reduction in surface mass balance (SMB), as well as dynamic factors; with meltwater runoff now being the main contributor to ice loss (Mouginot et al., 2019; van den Broeke et al., 2016; The IMBIE Team, 2020). Increased surface melt, driven by atmospheric warming, causes a direct reduction in mass through surface runoff, as well as mass loss through ice-dynamical feedback processes (e.g. Bartholomew et al., 2012; Zwally, 2002)





Since the early 1990's, the GrIS has experienced a summer temperature increase of ~1.7°C (Hanna et al., 2021), with a
commensurate increase in surface meltwater production through an increase in melt extent as well as enhanced local melt rates
(Hall et al., 2013; van As et al., 2012), and surface runoff has risen by 33-50% since the early 2000s (Trusel et al., 2018).
Surface melting on the GrIS has migrated to higher elevations since 2000 (Gledhill and Williamson, 2018), and meltwater
features (e.g., lakes, rivers and slush) have followed (Howat., et al., 2013; Tedstone and Machguth, 2022). This trend is
expected to continue as the climate warms further - temperatures are predicted to increase by up to 6.6 °C by 2100 (Hanna et
al., 2021) - with models suggesting that meltwater features will extend 110 km further inland by 2060 under extreme
warming/emission scenarios (Leeson et al., 2015).

Surface melt on the GrIS can pond in supraglacial lakes (SGLs), flow in rivers/streams, and also saturate snow and firn to
create surface slush. SGLs generally form in early summer enlarge in area and depth between spring and summer as they
accumulate water, and can drain either rapidly in hours by hydrofracture, slowly in days to weeks via channel incision and
overflow, or may instead refreeze at the end of the melt season (Box and Ski, 2007; McMillan et al., 2007; Sneed and Hamilton,
2007; Selmes et al., 2011; Williamson et al., 2017; Doyle et al., 2013). Moulins, often created by lake hydrofracture events
(e.g. Das et al., 2008; Tedesco et al., 2013), allow surface meltwater to access the ice sheet base, where the location and timing
of meltwater injection can modulate ice flow (Zwally et al., 2002; Bartholomew et al., 2010; 2012; Hoffman et al., 2011; Sole
et al., 2011; Nienow et al., 2017). Once moulins have opened, they may act as surface-bed connections for the remainder of
the melt season, enabling meltwater to impact ice dynamics over monthly-to-seasonal timescales (Joughin et al., 2008, Banwell
et al., 2013, 2016; Hoffman et al., 2018). Meltwater can also travel laterally across the ice sheet surface through slush fields,
which are features that have been identified on the ice sheet as early as the 1950s (Holmes, 1955). Slush fields can initiate or
reopen supraglacial channel routes (Miller et al., 2018; 2020; Machguth et al., 2023; Clerx et al., 2022). Refrozen slush and
SGLs can create impermeable ice slabs, which inhibits water storage in the underlying firn and, in turn, increases ice surface
runoff and ultimately contributes to global sea-level rise (MacFerrin et al., 2019; Tedstone and Machguth, 2022).

As meltwater runoff is the main contributor to GrIS mass loss (The IMBIE Team, 2020), the distribution of surface meltwater
on Greenland has been the focus of several modelling (e.g. Banwell et al., 2012; Leeson et al., 2012) and remote-sensing
studies (e.g. Yang and Smith., 2013; McMillan et al., 2007; Selmes et al., 2011; Williamson et al., 2017, 2018a; Miles et al.,
2017; Yang et al., 2021). Previously, the characteristics and behaviour of medium to large SGLs (i.e., those > 0.0495 km²)
have been studied in single melt-seasons (e.g. Williamson et al., 2017, 2018a; Miles et al., 2017). Although several multi-
seasonal studies have been conducted, they have often been limited by coarse spatial resolution data and SGLs < 0.125 km²
have been ignored (e.g. Selmes et al., 2013; Fitzpatrick et al., 2014). This is often justified by the fact that large volumes of
water are needed to hydrofracture through ~km thick ice (Krawczynski et al., 2009), however since small lakes tend to form
at the ice margin where the ice is thinner, it is possible that they may trigger hydrofracture and perturb ice dynamics there
(Williamson et al., 2018a).



Even though supraglacial meltwater on the GrIS exists in different forms, previous remote sensing studies have tended to focus
on lakes, rivers and slush as separate entities (e.g. Box and Ski, 2007; Selmes et al., 2011; Williamson et al., 2017; Sundal et
al., 2009; Smith et al., 2015; Greuell and Knap, 2000;  Machguth et al., 2023; Tedstone and Machguth, 2022). Although there
have been a handful of studies that have focussed on mapping and analysing the supraglacial hydrological system as a whole
(e.g. Rawlins et al., 2023; Zhang et al., 2023; Yang et al., 2021), the drainage of meltwater features was not considered.
Likewise, some studies included subsurface lakes (e.g. Dunmire et al., 2021; Miles et al., 2017), although surface slush features
were ignored. Despite recent research, little is known about how the distribution of surface meltwater, including slush and lake
drainage events, differs as a whole system across fine spatial and temporal scales, and how this varies between high and low
melt years.

Since the distribution and dynamics of supraglacial meltwater has a profound influence on the mechanisms contributing to
mass loss from the GrIS, it is therefore important to understand its evolution, especially since high melt years - presumably
more conducive to melt and meltwater ponding - are becoming more frequent as the climate warms. In this study, we compare
the distribution and morphology of supraglacial meltwater features in the Russell/Leverett glacier catchment, southwest
Greenland, in the low melt year of 2018, to the extreme high melt year of 2019. Previous studies have focused on longer time
periods or larger areas, but at the expense of accuracy and, consequently, these studies omit meltwater features that have the
potential to play an important role in the supraglacial meltwater system. Here, we precisely delineate all surface meltwater
features (i.e SGLs > 0.0018 km², as well as rivers and slush) using normalised difference water index (NDWI) thresholding
methods and additional delineation applied to both Landsat 8 and Sentinel-2 optical satellite imagery. By undertaking this
detailed mapping, we can gain a fuller understanding of how the characteristics of the supraglacial hydrological network
change in extreme high and extreme low melt years, how a warming climate can impact meltwater characteristics and
dynamics, and also gain an insight to the potential implications of surface meltwater in the warming future.



**Figure 1: Maximum areal extent of supraglacial meltwater features (blue) in 2018 (a) and 2019 (b) within the Russell/Leverett**
**Glacier catchment (black outline). Elevation contours from the ArcticDEM are shown in grey. Background is a true colour Sentinel-2 image acquired on 26/09/2019. Inset depicts the location of the catchment within the southwest GrIS.**



## 2. Methods

### 2.1 Study area

We focus on a ~5800 km² area of Southwest Greenland: the Russell/Leverett Glacier catchment (Figure 1). The surface
drainage basin is derived from ArcticDEM Digital Elevation Model at 1 km resolution. The basin is land terminating and
meltwater is transported from the ice sheet margin oceanward by both Watson and Isortoq proglacial rivers. The study area is
well known for prevalent surface hydrology features including lakes, rivers, and moulins (e.g., Bartholomew et al., 2010;
Sundal et al., 2009; Smith et al., 2015; Fitzpatrick et al., 2014; Yang et al., 2021).

### 2.2 Study years

We focus on two melt seasons: the relatively low melt year of 2018, and the relatively high melt year of 2019. The 2018 melt
season was anomalously cold, with a summer mean temperature anomaly of up to -1.5 °C relative to 2002–2016 (Sasgen et
al., 2020). These cold temperatures coincide with a 65-78 Gt yr$^{-1}$ reduction in meltwater runoff in comparison to the 2003-
2018 mean (Sasgen et al., 2020). Surface melt extent in 2018 reached 44% of the total ice sheet area, only slightly above the
1981-2010 average of 40 % (Tedesco et al., 2019). Throughout 2018, ~100 Gt of mass was lost from the GrIS, a 58% reduction
compared to the 2003-2018 average of ~235 Gt yr$^{-1}$ (Sasgen et al., 2020).

In contrast to 2018, 2019 was exceptionally warm, with some regions reaching a summer mean temperature anomaly of up to
+ 1.5 °C (relative to 2002–2016, Sasgen et al., 2020). 2019 exhibited extremely high runoff rates due to the exceptionally
warm summer season, which neared the maximum rate between 2002-2018 set in 2012 (Sasgen et al., 2020). Surface melt
extent in 2019 reached a maximum of 60% of the total ice sheet area, greatly exceeding the 1981-2010 average of 40%
(Tedesco et al., 2019). The GrIS experienced record high rates of mass loss throughout 2019 when ~532 Gt was lost; more
than double that of the 2003 to 2018 average and 15% greater than the prior record high melt year of 2012 (Sasgen et al., 2020;
Mohajerani, 2020).

### 2.3 Satellite imagery and pre-processing

All available imagery from both Landsat 8 Operational Land Imager (hereby 'L8') and Sentinel-2 MultiSpectral Instrument
(S2) satellites was acquired from 1 May to 30 September for both 2018 and 2019 melt seasons (Table S1). Images with > 50%
cloud cover were omitted, and images with a sun angle < 20 ° were discarded due to difficulties in accurately differentiating
meltwater features from adjacent features under these conditions (Halberstadt et al., 2020). 10 L8 images and 18 S2 images
were used for 2018, and 16 L8 and 63 S2 images were used for 2019, corresponding to a mean temporal sampling of ~ 5 days
and ~ 2 days in 2018 and 2019, respectively.



L8 data were downloaded as level-1T geometrically and radiometrically calibrated images in the form of Digital Numbers. The L8 Level-1T data were converted to Top-Of-Atmosphere (TOA) reflectance using individual image metadata and equations provided in the Landsat 8 Data Users Handbook (USGS, 2019). L8 bands 2 (blue), 3 (green), 4 (red) and 5 (NIR),

which have a spatial resolution of 30 m, were pan-sharpened to a 15 m resolution using intensity hue saturation methods, Rahmani et al., 2010) to better match the 10 m resolution of S2 data. L8's band 6 (SWIR) and 10 (thermal infrared), which have a spatial resolution of 30 and 100 m, respectively, were resampled using nearest neighbour interpolation. S2 data were downloaded as Level 1-C orthorectified TOA reflectance products with sub-pixel multispectral registration. S2 Bands 1 (Coastal Aerosol) and 11 (SWIR) have a spatial resolution of 60 and 20 m, respectively, and so were resampled using nearest

neighbour interpolation to match the finer (10 m) resolution of bands 2 (blue), 3, (green), 4 (red) and 8 (NIR).

### 2.4 Supraglacial meltwater delineation

We delineate surface meltwater features in L8 and S2 imagery using methods by Corr et al. (2022), which are summarised here and in Figure 2, where all threshold values are also stated. Prior to meltwater delineation by Normalised Difference Water Index (NDWI) thresholding, rock and cloud masks are created and applied to each L8 and S2 image to reduce classification

errors, following Corr et al. (2022) (Figure 2). The addition of masks allows for better separation between deep meltwater features, rocks, clouds, and shaded snow areas. For L8 images, rock masks were created using the thermal infrared band, blue band, and red band. For S2 imagery, rock masks were created by applying the Normalised Difference Snow Index (NDSI), created by Hall et al. (1995), as well as an additional blue and green filter. Clouds were masked from L8 imagery using the NDSI, Short-Wave Infrared band (SWIR), and blue band. Clouds were masked in S2 imagery using the SWIR band, SWIR-

Cirrus band, and blue band. We take the threshold values for these masks directly from Corr et al. (2022) (Figure 2).

To delineate the meltwater features in each image, we apply the NDWI, calculated from the ratio of green and NIR wavelengths, in addition to the NDWI adapted for ice (NDWIice), which utilises the ratio of blue and red wavelengths and better accounts for the surface conditions of the GrIS (Yang and Smith, 2013). For both L8 and S2 imagery, NDWI values >

0.24 and NDWIice values > 0.25 were classified as water pixels with all other pixels designated as not water (Figure 2). Our threshold values are lower than those used by Corr et al. (2022), which was a deliberate choice as we wanted to detect shallower meltwater features (including small streams and slush) than those considered in that study. Our chosen threshold values are in line with those from previous studies, which typically lie between 0.15 and 0.30 (e.g., Williamson et al., 2018a; 2018b; Miles et al., 2017; Bell et al., 2017; Stokes et al., 2019; Yang and Smith, 2012). For both L8 and S2 imagery, after application of the

NDWI and NDWIice, we implemented two further threshold values to better distinguish meltwater features from surrounding ice/snow: the green band subtracted by the red band, and the green band subtracted by the blue band, again using the values from Corr et al. (2022) (Figure 2).

After meltwater features had been delineated, binary water/non-water masks were created from all L8 and S2 scenes. In line
with similar studies (Stokes et al., 2019; Dell et al., 2020; Langley et al., 2016), meltwater features ≤ 1800 m² (≤ 2 L8 and ≤
18 S2 image pixels) were removed from the binary meltwater images to account for misclassification errors. Binary meltwater
images were then converted to polygon features. After NDWI thresholding, automatically identified features were then
partitioned into slush and rivers by manual interpretation based on the geometry and colour of the features. Slush was identified
as large areas of dense, light blue meltwater with poorly defined boundaries (e.g. Rawlins et al., 2023). Rivers/streams were
identified as narrow, linear meltwater features.

To manage detection of false positives and/or negatives in the polygon features, manual enhancement was done for all image
acquisitions used in the study by comparing the appearance of surface meltwater on true colour composite images. Polygons
which incorrectly identified surface water were manually removed, and undetected meltwater features such as deep lake centres
and narrow channels were manually added (Table S2). The uncertainty associated with using a dual sensor approach is
generally low, with the greatest uncertainties observed during peak season, with an $R^2$ of 0.93 and RMSE = 0.1 km²
(Supplementary text S1; Figure S1).

In the absence of extensive ground truth data, it is very difficult to assess the accuracy of lake area estimates, and this is an
acknowledged challenge in the literature (e.g., McMillan et al., 2007; Sundal et al., 2009; Leeson et al., 2012; Corr et al.,
2022). To provide an indicative estimate of the confidence in our manual delineation, we therefore created a ½ pixel width
buffer around the periphery of ~750 meltwater features, and by adding and subtracting this buffer assessed the proportional
impact upon our lake area estimates. This analysis indicates that if we systematically over- or under-estimate lakes by ½ a
pixel, then the expected error in lake area would be ~12%. In reality, we expect this to be a conservative (upper bound) estimate
of the true uncertainty, because it assumes that we systematically over- or under-estimate the area around the entire boundary
of a lake.

To determine meltwater feature characteristics from each meltwater feature polygon, the mean elevation and surface gradient
of each individual waterbody polygon were extracted from the ArcticDEM at 100 m spatial resolution (Porter et al., 2018).
Mean ice thickness values for every meltwater feature were extracted from BedMachine Greenland v4 at 150 m spatial
resolution (Morlighem et al., 2017; 2022).



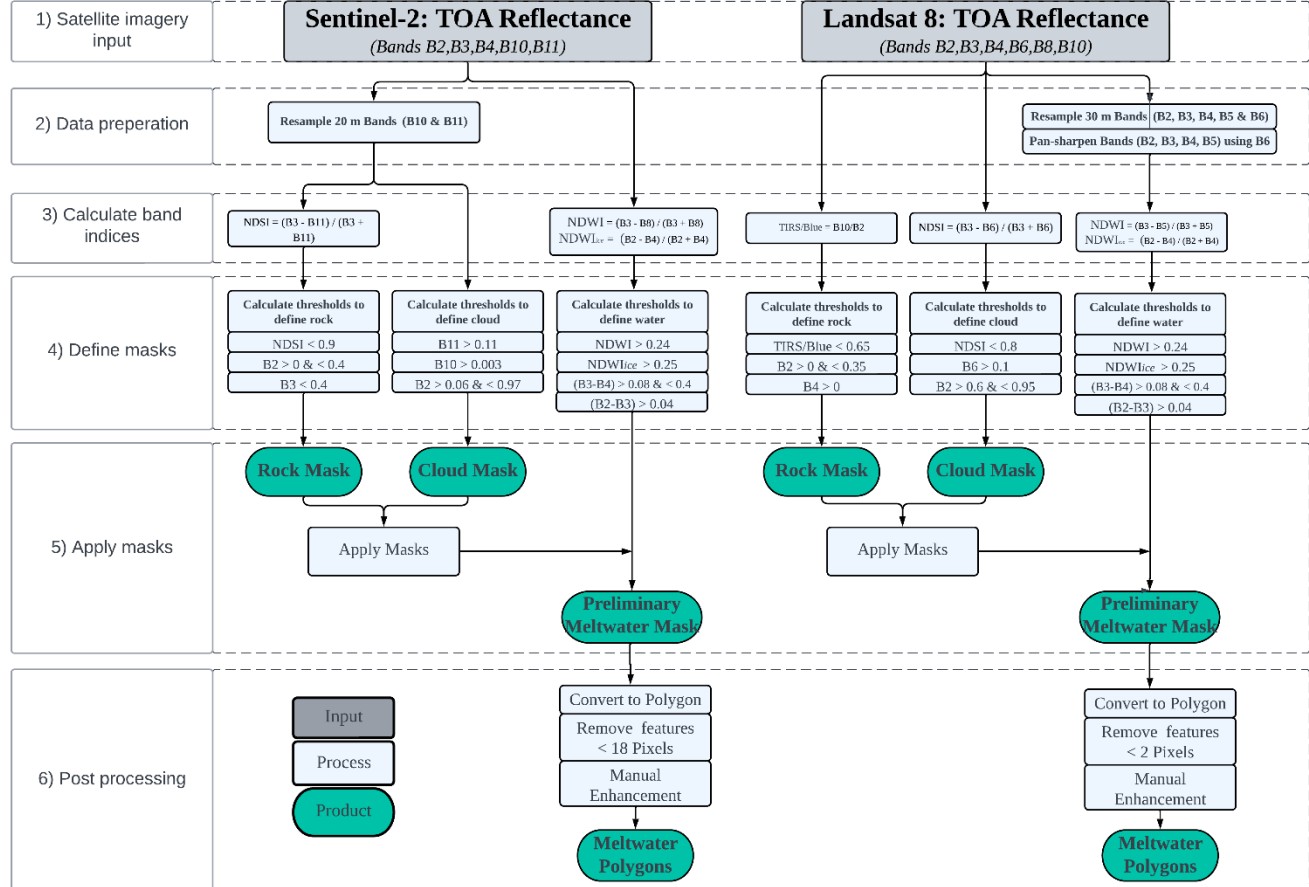


**Figure 2: Workflow of meltwater feature delineation by NDWI thresholding and additional filtering on Sentinel-2 (left) and Landsat 8 (right) image scenes, following methods adapted from Corr et al. (2022). Top of Atmosphere (TOA) reflectance values are used as input. Normalised Difference Snow Index (NDSI), Normalised Difference Water Index (NDWI) and additional band (B) combinations are used to mask imagery and define meltwater. The wavelength of each numbered band is in line with both the**
**Sentinel-2 User Handbook and Landsat 8 Data User Handbook (see Data Availability section).**

## 2.5 Meltwater feature depth and volume

Water depth for each meltwater feature pixel was determined using the physically-based radiative transfer model used in a variety of prior studies (e.g., Sneed and Hamilton, 2007; Banwell et al., 2014; Pope et al., 2016; Williamson et al., 2018a;
Macdonald et al., 2018). This algorithm calculates meltwater feature depth assuming that light penetrating a water column is attenuated with depth (Philpot, 1989). An assumption is made that the optical properties of the meltwater features are not altered by wind-driven surface roughness or column-integrated particulate matter. The lake bottom albedo is also taken to be homogenous (Sneed and Hamilton, 2007). Depth, z, is calculated from equation 1:




$$z = \frac{\ln(Ad - R\infty) - \ln(Rw - R\infty)}{g},$$

(1)

where Ad is lakebed reflectance, R∞ is the reflectance of optically deep water, Rw is the satellite measured reflectance value

of a water pixel, and g is the attenuation coefficient rate, which is associated with losses in upward and downward directions

through the waterbody.

For L8 data, depths were determined by an average of both red and panchromatic TOA reflectance values (after Williamson

et al., 2018a, Macdonald et al., 2018, and Pope et al., 2016). Ad was obtained from the average reflectance of the first non-

water pixel surrounding each feature, determined by a single pixel (30 m) buffer around meltwater features. R∞ was determined

in each individual image as the darkest open ocean pixel and if deep water was not present in an image, R∞ was taken to be 0

(Banwell et al., 2019; Dell et al., 2020). Following Williamson et al., (2018a) the value of g was taken to be 0.3817 for the

panchromatic band and 0.7504 for the red band (after Pope et al., 2016).


For S2 data, depths were determined by using the red TOA reflectance value, after Williamson et al. (2018a). Due to S2's finer

spatial resolution compared to L8, Ad was determined by the average reflectance of the first three pixels surrounding the

waterbody, as opposed to just one surrounding pixel for L8. R∞ was taken to be 0, as for L8 (see above), and the g value of

0.8304 was taken from Williamson et al. (2018a).


For both the L8 and S2 imagery, meltwater feature volumes were then calculated by taking the sum of depths multiplied by

the pixel area. We assume an uncertainty of 21.2% on these volume estimates, after Melling et al., (2023) who compared

meltwater feature depth and volume determined by the same radiative transfer algorithm to those determined from the

ArcticDEM Digital Elevation Model, for five lakes in the Russell/Leverett catchment.


## 2.6 Tracking meltwater features

The seasonal evolution and drainage dynamics of meltwater features within the catchment were tracked using the Fully

Automated Supraglacial lake Tracking at Enhanced Resolution (FASTER) algorithm developed by Williamson et al. (2018a).

Using this algorithm, we created maximum meltwater extent array masks for both the 2018 and 2019 melt seasons by

superimposing individual lake masks derived from each image. We then used FASTER to track meltwater feature area and





volume by detecting changes between subsequent imagery within each mask, which also enabled us to detect meltwater feature drainage events.

Williamson et al. (2018a) also used the FASTER algorithm to partition between rapid and slow lake drainage events based on
a 4-day sampling period. However, given the paucity of our data in 2018 relative to 2019, we performed an assessment of the temporal sampling that would be required in order to robustly determine whether a meltwater feature drained rapidly or slowly (Supplementary text S2; Figure S2). Our findings show that data with a temporal sampling of two to three days is required for such partitioning, while the typical temporal sampling of data in 2018 is only five days. As such, in this study, we restrict our analysis to assessing drainage of any kind, regardless of timeframe, vs. refreezing.


A meltwater feature was determined to have drained or re-frozen when > 20% of its liquid volume appeared to have been lost and if it did not gain volume in any subsequent image. This was subsequently partitioned between drainage and refreezing using RACMO air temperature data (see section 2.7). To determine the occurrence and timing of refreezing, we extracted temperature values from the centroid point of each meltwater feature during and two days preceding the volume loss event.
Following Selmes et al. (2013), we assumed that for a refreezing event to occur, the average mean air temperature had to be < 0°C over this period.

## 2.7 Regional air temperature and melt

To calculate temperature anomalies for the Russell/Leverett Glacier catchment, we use daily catchment averages of 2 m air temperature from the Regional Atmospheric Climate Model (RACMO2.3p2) at 5.5 km spatial resolution and further
statistically downscaled to a 1 km grid, hereafter 'RACMO'(Noël et al., 2018; 2019). We compare the daily mean temperatures throughout the 2018 and 2019 melt seasons with the mean temperature for each day of year for all years between 1958 and 2017. We also extract daily catchment averages of total melt (ice and snow) from RACMO throughout both 2018 and 2019 melt seasons.

## 2.8 Ice velocity

Ice velocity measurements were acquired from NASA MEaSUREs ITS_LIVE data cubes via the Global Glacier Velocity Point Data Access portal (Gardner et al., 2023). For the durations of both the 2018 and 2019 melt seasons, velocity values were acquired from image-pairs separated by 10 days at 10 points along elevation contours at 800 m, 1200 m, 1600 m and 2000 m. A separation period of 10 days was selected as we required a high temporal resolution to resolve velocity response to lake drainage, but also required sufficient coverage over the season. A three-day smoothing window was used to reduce noise.




**2.9 Proglacial discharge**

To infer ice-sheet meltwater runoff variability from the Russell/Leverett glacier catchment, we obtained in-situ daily measurements of meltwater discharge in both 2018 and 2019 from 1 May to 30 September through the Watson River, monitored by GEUS under the Programme for Monitoring of the Greenland Ice Sheet (PROMICE) and downloaded from the GEUS Dataverse (van As et al., 2018; 2022).

**3. Results**

**3.1 Distribution and morphology of meltwater features**

Throughout the 2018 and 2019 melt seasons, we identify 580 and 1410 active meltwater features, which cover 1.7% and 4.3% of the total catchment area over the course of the melt season, respectively (Figure 1). Meltwater extends from the margin to 265 km and 315 km inland in 2018 and 2019, reaching elevations of 1800 and 2000 m a.s.l., respectively.


In 2018 and 2019, 83% and 94% of meltwater features have an area < 0.0495 km² and are thus classed as small features. In both years (Figure 1), small features tend to form at the start of the melt season at low elevations, though are more abundant in 2019. In 2019, 90% of meltwater features that form within the first month of the melt season (May) are small and are situated at a mean elevation of 950 m a.s.l. In 2018, small features make up 87% of all features that form within the first month of the

melt season (June), and are situated at a mean elevation of 1040 m a.s.l. In both years (Figure 1), meltwater features tend to grow larger at higher elevations, coincident with lower surface slopes and thicker ice. In 2018 and 2019 respectively, features < 1200 m a.s.l. have a mean area of 0.043 km² and 0.024 km² compared to 0.047 km² and 0.042 km² at elevations > 1200 m a.s.l.

In 2019, linear stream and river features can be identified across the entire catchment, whereas in 2018 we tend to observe such features only at elevations > 1000 m a.s.l (Figure 1). In 2019 widespread slush patches (i.e., patches of dense, light blue meltwater with poorly defined boundaries which are clearly not lakes or streams) are observed at high elevations > ~1800 m a.s.l., whereas we observe very minimal slush in the 2018 melt season.

**3.2 Seasonal evolution of meltwater features**

During the 2018 melt season, meltwater features first appear in early June, and the cumulative area and volume of meltwater within the catchment increases (by 0.1 x 10$^7$ m² d$^{-1}$ and 0.09 x 10$^7$ m³ d$^{-1}$, respectively) throughout this month (Figure 3a). Total meltwater area and volume peak in mid-July 2018, reaching 3.7 x 10$^7$ m² and 4.2 x 10$^7$ m³, respectively. Both meltwater area and volume plateau throughout the remainder of July, but then abruptly decrease at the end of August, coinciding with





lower air surface temperatures (Figure 3a). Daily estimates of melt from RACMO indicate that the spatial mean of melt within the catchment fluctuates throughout 2018, but remains above zero until the end of August (Figure 3a).

During the 2019 melt season, meltwater features appear in early May, a month earlier than in 2018 (Figure 3). Total meltwater area remains low until the end of May 2019, then increases throughout June (by $0.2 \times 10^7$ m² d$^{-1}$) where it peaks at ($7.2 \times 10^7$

m²). Meltwater area subsequently declines until mid-July, but then rises abruptly to a second peak at the end of the month, reaching an areal extent ($8.3 \times 10^7$ m²) that is slightly higher than the first peak in mid-June. Meltwater area then progressively reduces until the end of the 2019 melt season. We note that the air surface temperature anomaly reaches a maximum of +10 °C, coinciding with both peaks in total areal meltwater extent in 2019 (Figure 3b). Meltwater volume increases gradually throughout May (by $0.03 \times 10^7$ m³ d$^{-1}$) and peaks in mid-June (at $7.3 \times 10^7$ m³). Volume declines throughout July, though with

a temporary increase at the end of the month. This increase in volume is notably smaller and shorter-lived than the increase in area at the same time, suggesting that meltwater features at this time are widespread but not very deep (with an average depth of 0.3 m on 20 June compared to 0.1 m on 1 August). From August onwards, meltwater volume steadily decreases throughout the remainder of the season (Figure 3b).



**Figure 3: Time series of total area (red) and volume (blue) of meltwater features in (a) 2018 and (b) 2019 from L8 and S2 imagery. Area (red) error bars represent a +/- 12% estimate of uncertainty and volume error bars (blue) represent a 21.6% estimate of uncertainty. Also shown is cloud cover percentage (black bars), modelled 2 m air temperature anomaly (green line) and spatial standard deviation (green shading), and total daily melt (mm w.e.; light blue line) and spatial standard deviation (light blue shading).**





### 3.3 Meltwater drainage dynamics

### 3.3.1 Drainage

Of the 580 and 1410 meltwater features that form in 2018 and 2019 respectively, 77% (466) and 92% (1294) are observed to drain (either rapidly or slowly). This corresponds to a similar total drained volume in each year (48.2 x $10^7$ m³ and 48.9 x $10^7$

m³ for 2018 and 2019, respectively), but the drained area in 2019 is over six times greater than that in 2018 (13.8 x $10^7$ m² and 2.2 x $10^7$ m², Table 1). Drainage begins in June in 2018 and May in 2019 and ends in August in both years.

In 2018, most of the meltwater volume and area drains in three distinct periods, occurring on 9 July, 6 August and 15 August (± 5 days; Table 1). The largest drainage event occurs on ~ 9 July, where 120 meltwater features drain, accounting for 46% (~

2.3 x $10^7$ m³) of total meltwater volume loss throughout the season. On ~ 6 August 2018, a total of ~25 meltwater features drain which contributes to 23% of total seasonal meltwater volume loss from the catchment. The cessation of drainage occurs in late August 2018, shortly after ~90 meltwater features drain on ~ 15 August.

In 2019, most of the meltwater volume drains throughout a two-week period in early June, accounting for 51% of total seasonal

meltwater volume loss. It is notable that in 2019, the majority of meltwater volume drains two months earlier than when most of the meltwater area decreases (Figure 5b), indicating that the majority of the meltwater volume is contained in few, deep, features in 2019 which drain early in the melt season. In 2019, the highest frequency of drainage events occurs in early August (a month later than 2018) where 50-175 meltwater features drain at high elevations (> 1800 m a.s.l), contributing only 1.6% to total season volume loss but 20% to the total season area loss (Figure 4b (iii); Figure 5). The last drainage event of 2019

occurs in mid-August, which is slightly earlier than in 2018.

In both years, early season drainage events tend to be small as, in 2018 and 2019, 87% and 94% of meltwater features that drain in the first month of each season have an area < 0.0495 km² (Figure 4). We identify that in 2018 and 2019, respectively, 44% and 36% of small drainage events occurred at regions where ice is < 1 km thick (i.e., at regions of thinner ice where a

much smaller volume of water for full thickness hydrofracture is required, Figure S3). We also find that in 2018 and 2019, 3.5% and 3.1% of the total meltwater volume drained through small meltwater features.

In both 2018 and 2019, drainage of meltwater features begin at lower elevations, and occur at increasing elevations/distance inland as the melt season progresses (Figure 4). The highest number of drainage events in 2019 occurs at high elevations -

1600 - 1800 m a.s.l., where 480 features drain, accounting for 35% of all events. In 2018, the highest frequency of drainage events occurs at considerably lower elevations of 1000 - 1200 m a.s.l., where 270 features drain, corresponding to 47% of all events (Figure 4a (iii)).



On 2 August 2019 (± 2 days), we note that a group of ~ 20 meltwater features drained in a short-lived event that we assume is
a result of cascading lake drainage (Christoffersen et al., 2018). This event occurred in the north of the catchment across
elevations from 800 to 1400 m a.s.l. (Figure 4b (i), at a region that overlays a subglacial trough network (Figure S3).

### 3.3.2 Refreezing

In 2018 and 2019, 23% (134) and 8% (116) of meltwater features refreeze corresponding to a total refreeze volume of 5.7 x
$10^7$ m³ and 11.6 x $10^7$ m³ (11% and 19% of total volume loss), and total refreeze area of 1.0 x $10^7$ m² and 2.3 x $10^7$ m² (31 %
and 14 % of total area loss), respectively.

In 2018, 11% of total meltwater volume refreezes (5.7 x $10^7$ m³), compared to 19% in 2019 (11.6 x $10^7$ m³). In both melt years,
surface refreezing occurs at  higher mean elevations than drainage; we observe refreezing at mean elevations of ~1450 m a.s.l.
in each year, which is ~400 m higher than drainage in 2018 and ~75 m higher than drainage in 2019 (Table 1). In 2018, there
is a clear separation of refreezing and drainage at ~1400 m a.s.l., with a lack of refreezing events at lower elevations and
limited drainage events at higher elevations (Figure 4a (i)). In 2019, refreezing typically occurs at elevations <1000 m a.s.l as
well as in the northern catchment area between 1600 - 2000 m a.s.l. (Figure 4b (i)). We identify that, in August 2019 22% of
all refreezing features are regarded as slush, which typically occurs at elevations > 1800 m a.s.l.

**Table 1: Statistics of draining and refreezing meltwater features in 2018 and 2019, where draining and refreezing features were
identified using FASTER (Williamson et al., 2018a) and subsequently partitioned using RACMO2.3p2 2 m temperature data (Noël
et al., 2019), with a threshold of < 0 °C  identified as refreezing. DOY is the 'day of year' in 2018 and 2019. DOY sampling is
calculated by averaging the start drainage DOY and the end drainage DOY.**

| | 2018 | | 2019 | |
|---|---|---|---|---|
| **Statistic** | *Drainage* | *Refreeze* | *Drainage* | *Refreeze* |
| Number of events | 466 | 134 | 1294 | 116 |
| Mean event day of year | 194.2 | 200.3 | 178.3 | 180.2 |
| Mean DOY sampling (± days) | 5.8 | 6.0 | 2.2 | 4.3 |
| Mean volume loss (x$10^5$ m³) | 10.4 | 4.3 | 3.8 | 10.0 |
| Total volume loss (x$10^7$ m³) | 48.2 | 5.7 | 48.9 | 11.6 |
| Mean area loss (x$10^4$ m²) | 4.7 | 7.6 | 10.7 | 19.6 |
| Total area loss (x$10^7$ m²) | 2.2 | 1.0 | 13.8 | 2.3 |
| Mean elevation (m) | 1219 | 1464 | 1384 | 1454 |
| Mean surface slope (°) | 0.68 | 0.43 | 0.63 | 0.53 |




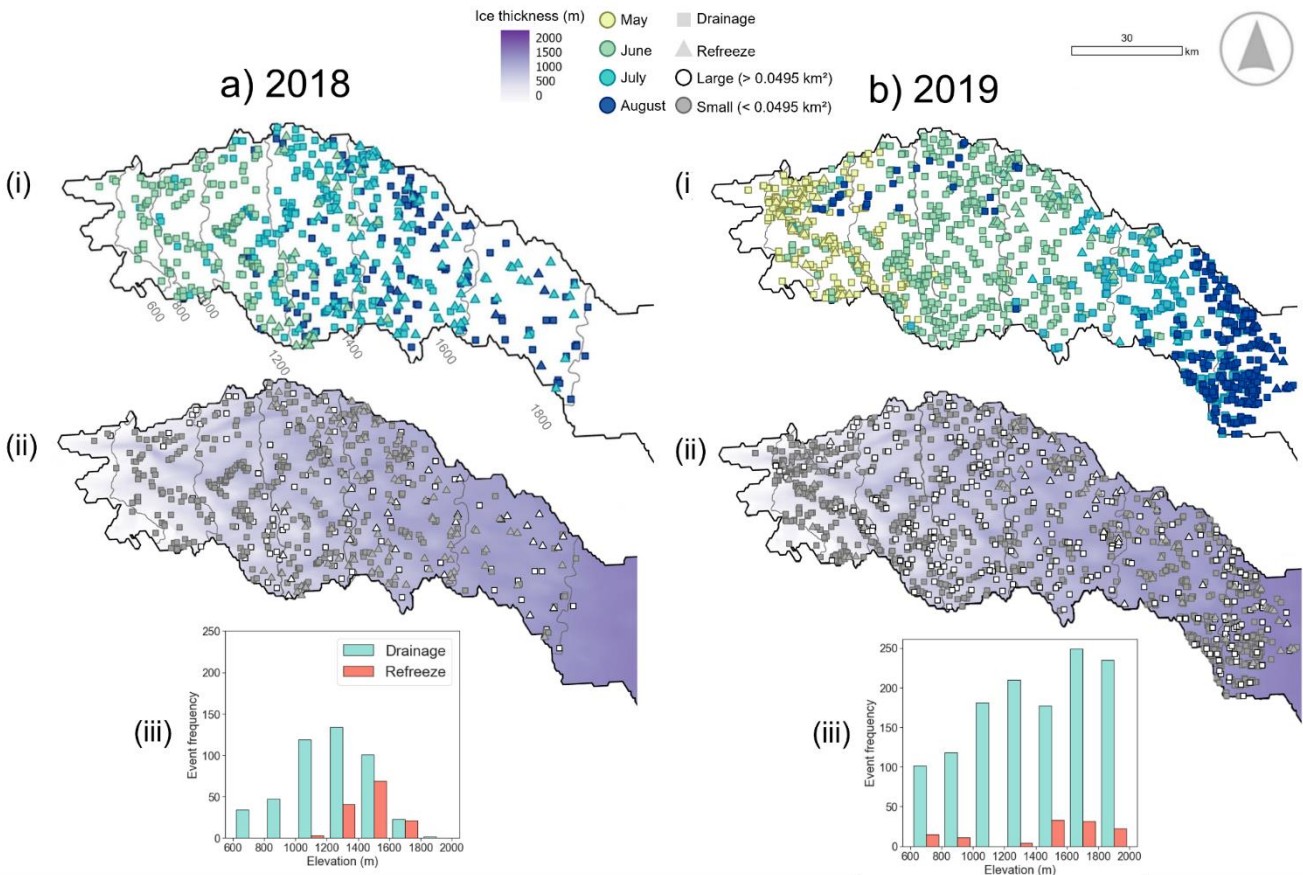

**Figure 4: Drainage and refreezing within the Russell/Leverett glacier catchment in 2018 (a) and 2019 (b). a(i) and b(i) depict the**
**timing of drainage (square) and refreezing (triangle) events in 2018 and 2019, respectively. a(ii) and b(ii) depict small (<0.0495 km²;**
**grey) and large (>0.0495 km²; white) drainage (square) and refreezing (triangle) events in 2018 and 2019, respectively. Light to dark**
**purple gradient represents ice thickness in metres. a(iii) and b(iii) depict drainage (blue) and refreeze (orange) events by elevation**
**in 2018 and 2019, respectively.**

### 3.4 Influence on ice velocity and pro-glacial discharge

The first instance of drainage observed in both years coincides with temporary increases in ice velocity. In 2018, this occurs
at lower elevations (Figure 5a (i)), where ice velocity increased by 0.1 km yr$^{-1}$ at elevations < 1200 m a.s.l. from 19 June - 26
June. As small meltwater features drained in early to mid-May 2019 (Figure 4b), velocity temporarily increased by ~ 0.2 km
yr$^{-1}$ at elevations between 1200 and 1600 m a.s.l (Figure 5b (i)). Discharge from the Watson River temporarily increases at the
same time as these small early season drainage events (Figure 5). In 2018, discharge increased by ~0.5 x 10³ m³ s$^{-1}$ as small
lakes drained. We also see two ~ 0.3 x 10³ m³ s$^{-1}$ temporary increases in river discharge in 2019 at the time of small lake
drainage.



On ~ 9 July 2018, we identify a temporary (~3 day) increase in ice velocity (of ~ 0.05 km yr$^{-1}$) at elevations of 1200 m a.s.l at the time of the largest drainage event in 2018 (Figure 5a (ii)). Discharge from the Watson River peaks after this speed up in

mid-July at ~ 1.5 x 10³ m³ s$^{-1}$ (Figure 5a). Between 29 July - 5 August 2018, ice velocity at 1600 m a.s.l increased (by ~ 0.2 km yr$^{-1}$), which coincided with a period of refreezing (Figure 5a (iii)). Ice velocity also gradually increases at 2000 m a.s.l at this time, which is sustained for ~ 3 weeks. A ~ 0.15 km yr$^{-1}$ increase in ice velocity occurs at elevations of 800 m a.s.l, occurring in between the drainage events on 6 August and 19 August (Figure 5a (iv)).

In 2019, when most meltwater drains in mid-June, mean ice velocity in the catchment rapidly increases at elevations > 1200 m a.s.l (Figure 5b (ii)). This increase is most apparent at high elevation regions, where we identify a 0.5 km yr$^{-1}$ speed up at 2000 m a.s.l which is sustained for ~ 2 weeks. Discharge from the Watson River gradually increases and is sustained at ~ 1 x 10³ m³ s$^{-1}$ at the same time. In early-August 2019, we see a ~ 0.1 km yr$^{-1}$ increase in velocity at low elevations of 800 m a.s.l. (Figure 5b (iii)), which coincides with the 2 August 2019 drainage event (Section 3.3.1). Discharge from the Watson River

also peaks at ~ 2 x 10³ m³ s$^{-1}$ in early-August, which occurs when slush is at its greatest extent (Figure 5b). Immediately after the drainage events in early-August, there is a transient ~ 0.2 km yr$^{-1}$ increase in ice velocity at 2000 m a.s.l. (Figure 5b (iii)), which occurs just after slush is at its greatest extent.

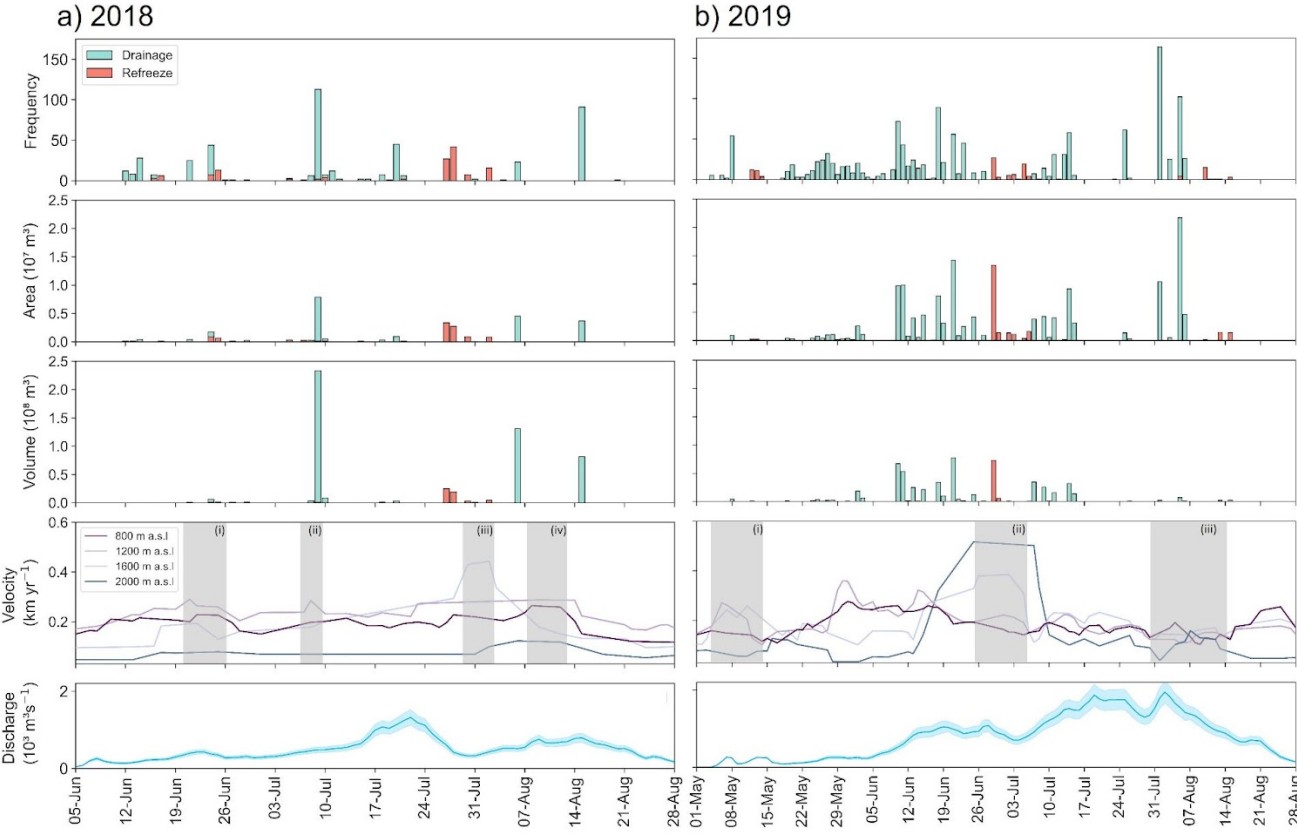

**Figure 5: Time series of meltwater feature drainage (green bar) and refreeze (orange bar) within the Russell/Leverett catchment in**
**a) 2018 and b) 2019. From top to bottom: frequency of drainage/refreeze events (i.e., the number of features that drained or refroze);**
**total daily area loss; total daily volume loss; ice velocity at 800 m a.s.l (dark purple), 1200 m a.s.l (light purple), 1600 m a.s.l (light**
**blue) and 2000 m a.s.l (dark blue) and velocity perturbations discussed in the text (vertical grey shaded columns, labelled i, ii, iii,**
**and iv); daily values of meltwater discharge through the Watson River (blue line) and associated uncertainty (blue shading). Note**
**that the x-axis ranges are different for each plot; each is constrained by the first and last meltwater feature drainage events in each**
**melt season.**

## 4. Discussion

This study identifies a clear contrast in surface meltwater characteristics and drainage dynamics between the high melt season
of 2019 and the relatively lower melt season of 2018. In 2019, surface meltwater features cover a much greater area and extend
further inland to higher elevations. This is in keeping with recent findings of Dunmire et al., (2021) and Zhang et al., (2023)
who also compared surface meltwater features within these two melt seasons. We find that meltwater features develop, and

total meltwater volume peaks about one month earlier in 2019 in comparison to 2018, which Rawlins et al. (2023) also identified on Humboldt Glacier in North Greenland for these same two melt seasons.

We have identified that a greater number, and larger total area, of meltwater features drained in 2019 compared to 2018, however our observations show that in both years a similar total volume drained. Since the volume of meltwater that drained in both years is similar, this may imply that high or low melt years may not necessarily influence seasonal dynamics. However, this finding may be due to limitations in the way volume is calculated using the radiative transfer method as deep meltwater features are not accurately resolved (Williamson et al., 2018b; Melling et al., 2023). Our observations show that in 2018 most meltwater drains in three discrete large scale events whereas in 2019, the majority of meltwater drains in multiple smaller

events but within a two-week period. However, we note that our 2018 dataset has a temporal resolution of ~ 5 days compared to the ~2 day resolution in 2019 and so it is possible that in reality drainage events in 2019 were sampled more finely.

In both melt years, we identify that surface meltwater features more commonly drain than refreeze, and that refreezing occurs at higher elevations than drainage. This is in keeping with the findings of Johansson et al. (2013) and Selmes et al. (2013). We

note however that it is difficult to distinguish between surface meltwater that has refrozen and that is stored as liquid in subsurface lakes (Dunmire et al., 2021, Miles et al., 2017). Dunmire et al. (2021) found that, in southwest Greenland, throughout the 2018 and 2019 melt seasons, 87 and 80 subsurface lakes were identified, respectively and these features were likely classified as refreezing events in our study.

More specifically, we observe differences between small meltwater features, high elevation meltwater features, the areal extent of slush and the potential occurrence of cascading drainage in particular between the two melt years. We discuss these findings in the following sections.

### 4.1 Small meltwater features

Small meltwater features (which we define as features ≥ 0.0018 and ≤ 0.0495 km²) are abundant and tend to form, and drain,

earlier in each melt season and at lower elevations than larger features. This is likely because more prevalent crevassing at lower elevations (Das et al., 2008) and smaller surface undulations (Johansson et al., 2013) inhibits their growth.

We find that these small lakes drain in both 2018 and 2019, although this behaviour is more prevalent in the high melt year of 2019. We identify that small meltwater features drain immediately prior to the onset of a temporary increase in ice velocities

at elevations < 1200 m a.s.l in 2018 and between 1200 and 1600 m a.s.l in 2019. Thus our observations suggest that such features are important in modulating short-term ice dynamics here, presumably because they drain early in the season when the subglacial hydrological network is inefficient (Bartholomew et al., 2010). The direct influence of the observed early season drainage of small meltwater features on ice velocity is short lived, likely due to the rapid development of efficient subglacial



hydrology. However, it is reasonable to assume that moulins are created when small meltwater features drain, which may then

modulate ice dynamics on seasonal time scales by providing access points for meltwater runoff to reach the ice sheet base throughout the remainder of the season (Banwell et al., 2016).

We note that we implement a lower minimum area threshold than other studies; for example, Williamson et al., (2018a) and Miles et al., (2017) restrict their analysis to lakes larger than 0.0495 km$^2$ on the basis that lakes smaller than this are unlikely

to trigger hydrofracture through 1 km thick ice (Krawczynski et al., 2009) and thus exert a control over ice flow. However, in our study we observe that - in both melt years - over a third of small lakes drained where ice is considerably thinner than 1 km thick, and thus it is likely that these lakes did in fact drain by hydrofracture. This suggests that moulin density at these elevations is likely to be higher than previously reported (Banwell et al., 2016; Hoffman et al., 2018). This is important because moulins created by 'small' lake drainage events route the majority of surface meltwater to the base of the ice sheet (Williamson et al.,

2018a). Including lakes smaller than 0.0495 km$^2$ in future studies is therefore essential if we are to effectively characterise, where, when and how much surface meltwater is routed to the ice sheet bed and any corresponding impact on ice flow.

## 4.2 High elevation meltwater features

We identify that meltwater features tend to form and drain at higher elevations in the high melt year of 2019 than the low melt

year of 2018. This is in keeping with previous studies which found that meltwater features tend to reach increasingly higher elevations during more intense melt-years (Sundal et al., 2009; Liang et al., 2012; Luthje et al., 2006). It is notable that even in the comparatively low melt year of 2018, the maximum elevation of meltwater features reached >200 m higher than the 2000-2010 average identified by Leeson et al. (2015). This is likely because higher melt occurred in 2018 than the 2000-2010 average, especially towards the end of the season when we note that high elevation meltwater features are more likely to grow.


We identify a transient increase in ice velocity at high elevations (2000 m a.s.l) after a series of drainage events at the same elevation in early August 2019. This suggests that the subglacial hydrology of the ice sheet is inefficient at these elevations and is in line with the findings of Fitzpatrick et al. (2013) and Doyle et al. (2014) who observe melt induced ice-speed up at 1200 m a.s.l (between 2009-2010) and 1800 m a.s.l (between 2008-2013), respectively, in the same glaciological catchment

as our study. Since high melt-years - and thus lake formation and drainage at high elevations - are likely to become more frequent in future years, meltwater-induced speed up events such as that reported here may occur more often in the future, potentially increasing ice flux to the margin and contributing to sea level rise.





### 4.3 Slush

We find that slush is far more widespread in the high melt year of 2019 than in the lower melt year of 2018, which we suggest is due to high rates of surface melt in the 2019 season above the snowline, where water is able to saturate surface snow and firn. Previous supraglacial hydrology studies focused on the GrIS have typically disregarded slush - potentially since it may have been scarce until recently.

Our observations show that the drainage of slush at high elevations coincides with a temporary increase in ice velocity at the same high elevation region. One possible mechanism for this may be due to the lateral and downslope supraglacial flow of water stored in slush towards high elevation moulins/crevasses, which we suggest formed by high elevation hydrofracture events (see section 4.2). We note that we do not see an increase in velocity at lower elevations, despite the fact that the abundance of slush likely indicates strong wide-spread melt at this time, likely because the subglacial hydrological system is

efficient here, and evacuates excess meltwater towards the ice sheet margin, before it can cause without which limits subglacial pressurisation and ice-speed up caused by meltwater accessing the base. We also note an increase in discharge from the Watson river when slush is at its greatest extent in August 2019, this is likely a result of strong wide-spread melt across the entire catchment at this time, augmented with additional meltwater previously stored in slush draining to the bed, and subsequently emerging as proglacial outflow.


   We identify that, in 2019, over double the area of meltwater refreezes compared to 2018, and this occurs at high elevation regions where slush is present. The refreezing of slush in high melt years like 2019 is likely to have a long term influence on the runoff from the GrIS. Slush has the ability to create impermeable ice slabs when it refreezes, creating an impermeable barrier for meltwater to access the underlying firn in subsequent years - ice slabs that formed in Greenland's interior after the

high melt season of 2012 persisted for five years (Culberg et al., 2021). As high melt years like 2019 become more frequent, it is likely that slush will become more prevalent, leading to formation and expansion of ice slabs and thus the preconditioning of the ice sheet surface for ponding in future years. This is potentially already occurring at Humboldt Glacier in North Greenland where studies have shown earlier activation of the hydrologic system and longer melt-seasons in years following widespread slush events (Rawlins et al., 2023).


### 4.4 Cascading drainage events

In 2019, a distinct cluster of ~ 20 lake drainage events occurred in August, at lower elevations than expected based on the drainage pattern of other lakes in our sample (Figure 5, Figure S3). These features were situated close together towards the North of the study area, and aligned over a subglacial trough. We suggest that these features drained via a chain-reaction style

'cascading drainage' process, whereby the rapid drainage of one meltwater feature induces surface strain rate perturbations





which trigger other surface meltwater features in close proximity to drain (Christoffersen et al., 2018). As more surface meltwater features drain, meltwater delivered to the base allows for basal sliding, ice speed up, crevasse formation and subsequent meltwater feature drainage upstream. This is supported by the fact that we also observe ice velocity increases at low elevations (800 m a.s.l) at this time, and that we would expect lakes aligned along a subglacial trough to be connected

through the subglacial hydrological network (Lindbäck et al., 2015). This is interesting because we would expect the subglacial hydrological network to be efficient by August, especially at lower elevations, and thus to effectively evacuate meltwater from a draining lake without causing an increase in effective pressure with an associated speed-up. These findings then support previous studies suggesting that meltwater supply variability can lead to an increase in ice flow, even in regions where subglacial hydrology is already efficient (Schoof 2010).


We also speculate that larger-scale cascading drainage events occur in 2018. Our observations show that in 2018 most meltwater drains in short-lived large scale events, and that increased ice velocity episodes follow; a pattern also noted by Christoffersen et al. (2018) in 2010. We note however that the temporal sampling of our dataset in 2018 is sparse, which means that our large-scale drainage events could in reality be multiple smaller scale events, and thus more finely sampled data is

probably needed to confirm these findings.

## 5. Conclusion

In this study, we have compared supraglacial meltwater features in the Russell/Leverett glacier catchment in the high melt year of 2019, and the comparatively low melt year of 2018. We identify that a greater number of meltwater features form which cover a larger proportion of the study area in the high melt-year of 2019 compared to the low melt-year of 2018. We determine

that more features and a larger total area of meltwater features drained in 2019 compared to 2018. For both melt years, we find that meltwater features tend to drain rather than refreeze, and that refreezing occurs at higher elevations than drainage.

We show that small meltwater features are an important component of the glacio-hydrological system in both high and low melt years - though their effects are more profound in 2019. They induce a short-lived increase in ice flow when they drain in

both years, and have the potential to influence ice dynamics on seasonal time scales by creating new surface-to-bed conduits. We also find that meltwater features form and drain at higher elevations during the more intense melt-year of 2019, and that these drainage events also induce a transient increase in local velocity. This suggests that as 2019-like conditions become more common in a warming climate, we could see a more sustained speed-up in ice flow here.

We identify a potential cascading drainage event in early August 2019, which appears to perturb ice velocity at lower elevations. This is unexpected because we would expect subglacial hydrology to be efficient here, and thus to evacuate excess water from the ice sheet bed before it can cause an increase in sliding. We also identify that slush is much more widespread at

higher elevations in the high melt year of 2019, although we acknowledge that our sample size of two years is small. Future work should therefore focus on understanding the longer-term distribution and evolution of slush on the GrIS, especially given

that slush formation, drainage and refreezing has the potential to influence both ice dynamics and meltwater runoff.

As Greenland warms, exacerbated by Arctic amplification and climate/ice-sheet feedback processes, the frequency of high melt years like 2019 will likely increase. Our study suggests that previously poorly mapped and under-studied supraglacial hydrological features may have a larger impact on supraglacial runoff and ice flow, and thus on global sea level rise, in future

warmer melt years.

**Author contributions**

EG and AL conceptualised the research. AL, AB, JM and MM provided supervision. DC developed the code to delineate meltwater features. BN provided RACMO2.3p2 data. EG carried out the main body of work and drafted this paper. All co-authors contributed to manuscript editing.

**Competing interests**


The authors have no competing interests to declare.

**Data availability**

Meltwater feature shapefiles are available by request from the lead author (e.glen@lancaster.ac.uk). S2 data was sourced from the Copernicus Open Access Hub: https://scihub.copernicus.eu/. and L8 data was obtained from the United States Geological Survey (USGS) Earth Resources Observation Science (EROS) Centre: https://eros.usgs.gov. Sentinel-2 User Handbook available from: https://sentinels.copernicus.eu/web/sentinel/user-guides/sentinel-2-msi and Landsat 8 Data Users Handbook available from: https://www.usgs.gov/media/files/landsat-8-data-users-handbook. The full MATLAB source code for the

FASTER algorithm is available for download (Williamson, 2018b). RACMO2.3p2 data is freely available from the authors of Noel et al. (2019). Ice velocity data are available from NASA's Inter-mission Time Series of Land Ice Velocity and Elevation (ITS_LIVE) project (https://its-live.jpl.nasa.gov/#data).

**Acknowledgments**




EG and AAL received support from the Natural Environment Research Council (NERC) MII Greenland project under award NE/S011390/1 and from the European Space agency under the POLAR+ 4DGreenland project, ESA Contract No. 4000132139/20/I-EF. AFB received support from the U.S. National Science Foundation (NSF) awarded under 1841607 to the University of Colorado Boulder. MM was supported by the UK NERC Centre for Polar Observation and Modelling, and the
Lancaster University-UKCEH Centre of Excellence in Environmental Data Science. BN was funded by the Fonds de la Recherche Scientifique de Belgique (F.R.S.-FNRS).

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
