# Peer review of "A comparison of supraglacial meltwater features throughout contrasting melt seasons: Southwest Greenland"

_EGUsphere, 2024_

## Referee Comment (RC1)

**Review of Glen et al. "A comparison of supraglacial meltwater features throughout contrasting melt seasons: Southwest Greenland", submitted to The Cryosphere.**

*Andrew Tedstone, University of Fribourg/University of Lausanne*

**Summary of study**

This study maps the seasonal evolution of supraglacial meltwater features in a surface drainage catchment on the western margin of the Greenland ice sheet during two melt seasons by digitising the features from multispectral satellite scenes. Next, it seeks to attribute meltwater disappearance to either drainage or refreezing. Finally, it examines the case for links between surface meltwater drainage and basal sliding.

The purported main findings are that (i) surface meltwater features evolve depending on surface melt, draining from higher elevations in warmer summers and (ii) the drainage of these features, even very small ones, can lead to transient ice velocity responses, which may therefore be important to the future stability of ice flow in response to enhanced melting.

**Major comments**

My overall view is that this study is weak. The basic methodology is not novel: mapping of supraglacial meltwater features is well-established in the literature, indeed as highlighted by the references in the study. This would not be a problem if the rest of the analysis made solid contributions on top of this approach, but I did not find this to be the case. I am unconvinced by the partitioning between drainage and refreezing, while the links with catchment wide ice velocity are in principle novel but appear to suffer from some major methodological flaws which I fear are unfixable. Please find more reasoning for my conclusions below. I am sorry that I cannot be more positive about the manuscript at this time. I am open to discussion if the authors feel I have mis-interpreted aspects of the study.

**Originality/novelty**

The mapping of supraglacial meltwater features here uses established techniques which have been previously applied to a range of moderate and fine resolution imagery (non-exhaustively: Williamson et al., 2018; Smith et al., 2015). In this respect, I did not really learn anything new about the basic evolving pattern of supraglacial drainage in this area of the ice sheet that has not already been evidenced elsewhere.

As I indicated above, the link to ice velocity has the potential to be novel, as does the attribution to meltwater disappearance between refreezing and drainage. However, I have major concerns about the quality of these two parts of the analysis.

**Scientific quality/rigour**

Here I concentrate on the two areas of analysis which I found most problematic.

*(i) Ice velocities.* The authors employ the ITS_LIVE velocity dataset, which is derived from feature tracking of optical satellite imagery. The details provided in this study's methods are insufficient. For instance: what are the uncertainties/errors? Was any filtering (as opposed to smoothing) of the ITS_LIVE data carried out? 10 days is a very short baseline considering the expected ice displacement over this period in a "slow" land-terminating catchment and also in relation to the imagery pixel size, so is surely prone to high uncertainties.

Concretely, without these details I am especially concerned by the analysis of retrievals above ~1,200 m asl. Among the several studies which have looked in detail at feature-tracked velocity retrievals in this area (e.g. Tedstone et al., 2015; Williams et al., 2020; Halas et al., 2022), it is clear that retrievals become very sparse to non-existent above 1,200-1,400 m asl, even when employing higher-resolution Sentinel-2 data, owing to a lack of features which can be reliably tracked. Of course, these studies examined annual net ice flow, not intra-annual flow, so are not directly comparable, but provide a conservative sense of the coverage of feature-tracked retrievals in this area of the ice sheet (or possibly even optimistic given the way in which they mosaic several acquisitions together).

In this light I was very surprised to see "unvalidated" use of ITS_LIVE velocities all the way up to the 2,000 m contour. Using the ITS_LIVE Binder (https://mybinder.org/v2/gh/nasa-jpl/its_live/main?urlpath=lab/tree/notebooks), I took a look at a single point of these data through time at roughly this elevation:

[Figure]

It is clear that at high elevations the short-baseline retrievals are exceptionally noisy. Previous GNSS observations in this sector (e.g. Sole et al., 2013; Doyle et al., 2014) have shown that ice speeds above 1,500 m asl are less than c. 100 m/yr and show maximum daily velocities of up to max. 300 m/yr, generally occurring over periods less than 10 days (based on Sole Fig. 2). So, there are lots of velocity retrievals above which are simply not supportable by reference to previous ground observations.

To be satisfied that the analysis presented in the present study is appropriate, I would need to see evidence of: (a) a robust filtering approach to treat the abundant outliers (not just a boxcar moving window as currently used to smooth the data); (b) ideally, examination of the underlying signal-to-noise ratio (i.e. does the ITS_LIVE algorithm even support these velocities?); (c) an error budget/uncertainty analysis. (d) At the highest elevations, I suggest to go further, interrogating the velocity fields with reference to the underlying imagery, as I suspect that spurious cross-correlations are being identified associated with ephemeral slush fields, which are much less 'stable' than the ice-incised supraglacial channels found at lower elevations. Put simply, at the moment I do not believe the velocity analysis for elevations higher than ~1,200 m/yr.

*(ii) Surface hydrology partitioning.* Like with the ice velocities, my concerns particularly relate to higher elevations of the catchment. The study uses air temperatures to apportion the disappearance of surface meltwater into either drainage or refreezing. When meltwater disappears between two successive satellite acquisitions, if air temperatures were positive it is assumed to drain, whereas if they were negative it is assumed to refreeze. This is almost certainly overly simplistic. First, for instance, going all the way back to Holmes (1955), there is evidence that meltwater can continue to flow in open channels for up to two weeks after the end of surface melting. There is therefore a substantial lag between the onset of negative air temperatures and the freeze-up of the surface. Second, I suspect that there is some antialiasing of the evolution of surface drainage features with 'drainage'. This particularly concerns slush fields, which can either collapse/incise into more spatially discrete, efficient supraglacial channels – thereby presumably allowing water to be evacuated more quickly – or never evolve as far as incision, instead allowing water to continue to flow through the matrix. In this case, just because the water disappears from view, does not mean that all that water has refrozen. Instead, it may still be discharging via the sub-surface, as shown by Clerx et al. (2022), in a water table that is below the height of the snow surface.

Secondarily, I am concerned about the suitability of the water depth retrieval algorithm to drainage at higher elevations. These algorithms were developed on the basis of a solid ice substrate, which is often not the case above the ELA. In particular, slush fields are composed of a porous water-filled snow matrix perched on top of an ice slab formed of refrozen meltwater (see e.g. Clerx et al., 2022). Thus, neither are they spectrally similar to supraglacial ice-incised channels, nor do their depths correspond to an entirely liquid column. This makes the water volume retrievals from these features problematic.

My fixation with 'higher elevations' may seem pedantic, but the reality is that the vast majority of this study's geographic area of interest lies at these 'higher elevations', so my concerns are relevant to a large percentage of the catchment.

**Significance/impact**

Given my perspective above, in my view this study is not able to make impactful insights into the fate of supraglacial meltwater on the Greenland Ice Sheet.

**Presentation quality**

This is overall reasonable. The manuscript is mostly clearly written. I suspect it could be made more concise in places (i.e. length reduced). The figures are basically fine, apart from Figure 5, which presents ice velocities by elevation band but does not also segment the drainage/refreeze events/area/volumes by elevation band. This makes it very difficult if not impossible to independently verify the proposed links between surface hydrology "events" and ice velocity/basal sliding.

I note that some data are indicated as 'on request'. According to TC submission requirements, this is not acceptable:

> https://www.the-cryosphere.net/submission.html
>
> *Files for the review process & preprint posting*
>
> *After the manuscript registration, you are kindly asked to upload those files which are necessary for the peer-review process. The following files are required:*
>
> *…*
>
> *data sets, model code, video supplements, or other assets to your manuscript should be submitted to a reliable repository receiving a DOI, cited in your manuscript, and included in your reference list. Reviewers can then access these relevant sources;*

**Minor comments**

In light of my major comments, I have only a small number of minor comments at this point.

*Clarification of terminology:* particularly in the methods, in general 'surface meltwater features' is employed, but sometimes 'lakes' or 'slush fields' are used instead. This is particularly the case in sect. 2.6, which initially claims to track 'meltwater features' but then uses this term and 'lakes' interchangeably. See also L175, 'accuracy of lake area estimates' concerning delineations, was this actually only for lakes (and not also channels etc), and if so, why?

*L185:* surface gradients of meltwater features were retrieved from ArcticDEM. Presumably this is highly sensitive to whether those features were water-filled at the time of data acquisition for the DEM? More details are needed to assess whether this is a valid approach.

*Sect. 2.7, use of RACMO*: Given the high quality of in-situ AWS measurements on the K-Transect, it is valid to consider/state the performance of RACMO along this transect. Referencing should be sufficient.

*L389 and around:* references to panel a of Fig. 5, but this is to drainage only, without also referencing the velocities panel. Please improve.

*L540-1:* 'perturb ice velocity at lower elevations…this is unexpected'. I disagree. Other studies, for example Doyle et al. (2014), Ryan et al. (2024), show that transient velocity variations occur

whenever the subglacial drainage system's capacity is overwhelmed by the rate of meltwater supply. Rather than considering "drainage efficiency" to be an absolute quantity, consider it instead relative to antecedent and event melt supply.

**References not in original manuscript**

Doyle, S. H., A. Hubbard, A. A. W. Fitzpatrick, D. van As, A. B. Mikkelsen, R. Pettersson, and B. Hubbard (2014), Persistent flow acceleration within the interior of the Greenland ice sheet, *Geophys. Res. Lett.*, 41, 899–905, doi:10.1002/2013GL058933.

Halas, Paul, Jérémie Mouginot, Basile de Fleurian, Petra M. Langebroek, Impact of seasonal fluctuations of ice velocity on decadal trends observed in Southwest Greenland, Remote Sensing of Environment, Volume 285, 2023, 113419, https://doi.org/10.1016/j.rse.2022.113419.

Ing, R. N., Nienow, P. W., Sole, A. J., Tedstone, A. J., & Mankoff, K. D. (2024). Minimal impact of late-season melt events on Greenland Ice Sheet annual motion. *Geophysical Research Letters*, 51, e2023GL106520. https://doi.org/10.1029/2023GL106520

Tedstone, A., Nienow, P., Gourmelen, N. *et al.* Decadal slowdown of a land-terminating sector of the Greenland Ice Sheet despite warming. *Nature* **526**, 692–695 (2015). https://doi.org/10.1038/nature15722

Williams, J.J., Gourmelen, N. & Nienow, P. Dynamic response of the Greenland ice sheet to recent cooling. *Sci Rep* **10**, 1647 (2020). https://doi.org/10.1038/s41598-020-58355-2

---

## Referee Comment (RC2)

*The Cryosphere*
Anonymous Reviewer

**A comparison of supraglacial meltwater features throughout contrasting melt seasons: Southwest Greenland**

*Emily Glen, Amber Leeson, Alison F. Banwell, Jennifer Maddalena, Diarmuid Corr, Brice Noël, Malcolm McMillan*
* * *
**General comments:**

This manuscript investigates the seasonal evolution of supraglacial meltwater features in the Russell/Leverett glacier catchment, SW Greenland, with a focus on drainage distribution and characteristics from a low (2018) and high (2019) melt year and assess its implications, including ice velocity and potential drainage response in future warmer years. Of particular interest is the attempt by the authors to include small (i.e., <0.0495 km$^2$) and shallow meltwater features (e.g., slush), which can be overlooked in mapping studies, however important to consider. The authors use a pre-existing method (Corr et al., 2022) for all supraglacial meltwater feature extraction and then seek to partition these features into those that drain and refreeze, with links to ice velocity events.

The main findings are that (i) surface meltwater feature characteristics and distribution differ between a high (2019) vs low (2018) melt season, with meltwater features developing earlier (May) and occurring further inland (2000 m a.s.l) in the high melt season (2019); (ii); small meltwater features (<0.0495 km$^2$), predominately small SGLs, are important features of the system with their drainage prevalent at lower elevations and; (iii) the drainage of features, including those that are small, can generate an ice velocity response, with inference that a sustained speed-up in ice flow may occur at this catchment in the future.

Whilst I appreciate the effort put into the development of this study and manuscript, my view is that this study is not particularly novel in terms of the methods used (well-versed in literature) or the location of the study area in SW Greenland, which has been well documented and the focus of many supraglacial hydrology studies over recent years. Additionally, whilst you present the importance of small meltwater features (<0.0495 km$^2$), particularly small SGLs, and appreciate why they were included in terms of your findings (as agree, they are important!), I struggle with your use of the term *'all meltwater features'*. From Figure 1, it seems your river and stream network is lacking (even with the resolution of imagery used), with only larger portions of rivers extracted, often associated with either inputting or outputting an SGL (interpreted from Figure 1). Therefore I think your use of the term *'all meltwater features'* insinuates complete supraglacial drainage network maps and analysis (including smaller, shallower features such as the river and stream network) across the Russell/Leverett catchment, which is misleading as believe this is a basic supraglacial network at best. I therefore think there needs to be increased clarity of this or further mapping to be undertaken.

I am also unsure as to the method used for partioning meltwater that 'drains' and refreezes' as it has been shown in the literature that meltwater can stay active for much longer (i.e., weeks) post surface melt cessation. I think there needs to be some further clarification as to the simplicity of the method and acknowledgement that meltwater can linger for longer post surface melt cessation. Additionally, a figure showing this occurring in imagery (from your manual interpretation) may help you here, which could be placed in your supplementary information.

In addition, the link to ice velocity and drainage/refreeze events I believe, at times, is rather subjective. I provide further comments on these issues below.

**Major comments:**

*Feature extraction and terminology:*

In Figure 1, you provide your mapped supraglacial meltwater features for 2018 and 2019 (only figure presenting these in the main body and supplementary information, which is a shame), however it mostly looks like only SGLs and some slush have been captured, with the caveat of some larger, wider sections of supraglacial rivers (these networks look rather fragmented and incomplete). After searching one of the images used via Copernicus Browser (2019-07-25) and comparing with other studies in this area which have captured supraglacial features, there looks to be a number of rivers and streams not captured, or fragmented, by your method.

This is a shame as you state in your *introduction 'we precisely delineate all surface meltwater features (i.e SGLs > 0.0018 km², as well as rivers and slush)'* (Line 85) and within your methodology you state: *'Our threshold values are lower than those used by Corr et al. (2022), which was a deliberate choice as we wanted to detect shallower meltwater features (including small streams and slush) than those considered in that study'* (Line 150-153). However I do not see many small, shallow streams or continuous river channels captured (perhaps a few within the slush regions at most). It is my understanding that a significant part of this paper was to map *all supraglacial meltwater features*, in particular those of small, shallow size, across the catchment to examine feature distribution, evolution and low-high melt year comparison, with an emphasis on their importance albeit their size. I therefore think there are a number of things that need to be addressed in this manuscript and a few avenues which could be taken to improve the dataset and/or clarity in this paper:

1) As stated already, it looks like you have only captured segments of some of the primary (larger) supraglacial river channels within your drainage network, mostly where they input or output an SGL. Therefore, I think you need to make this clear in your introduction and/or methodology that you only partially-capture these types of channels (i.e., primary rivers) and not smaller stream-type networks (i.e., secondary tributary networks which are shallower and transient) and explain why (method limitation?). I think to aid this, it would help to give brief definitions of these differences as per the literature (e.g., Pitcher and Smith, 2019) in your introduction and then, throughout the manuscript, make sure you refer to these features as 'rivers', rather than interchangably between 'streams' or 'rivers/streams' to make it clear (as you have in some areas of text).

2) After performing NDWI and subsequent thresholding (via Corr et al., 2022), you state you partition your features into slush and rivers via manual interpretation of geometry and colour. I think an additional figure showing these partitioned features (lakes, slush and rivers) mapped would enable the reader to visually understand and assess their differences (i.e., the distribution of individual meltwater features, the overall collective drainage network characteristics and meltwater drainage behaviour – particularly that of smaller SGLs in lower elevations which are discussed) across the two melt years. This could be completed by including multiple mapped subsets in a new figure (within the results in either *Section 3.1* or *3.2*) showing the distribution and subsequent differences in features across selected dates and/or zoom in sections for both the 2018 and 2019 melt seasons.

3) Following on from my previous comment about an additional figure, this would also help show how the network evolves seasonally across the two distinct years. A particular interest, as you discussed, is the drainage of small meltwater features (particularly at lower elevations) and how slush develops in these two years. In terms of slush, so far, I can only see your higher-elevation slush in Figure 1 (from your maximum extent). It would be therefore interesting to see how slush develops during the melt season alongside your other features.

4) You could, time- and review-dependent, try to adapt your thresholding method or perform further manual delineation to capture more of the smaller network (i.e., a more complete supraglacial river and stream network) alongside your other smaller features (SGLs) to give a more hollistic view of supraglacial meltwater, and its drainage as a whole, in this catchment. Supraglacial rivers and streams can make up a large portion of the supraglacial network, and so are important to consider alongside your captured, smaller SGLs. This is a suggestion to uphold the use of '*all meltwater features' to* elevate the paper, however, I understand that this may be a considerable undertaking.

**Ice velocity:**

I have no problem with the use of NASA MEaSUREs ITS_LIVE data – it is a well-used and useful data resource. However, there is a lack of acknowledgement of the error and uncertainty regarding this data and implications this may have on your inferred results. For example a lack of error envelopes for your ice velocity data presented in Figure 5 (particularly important for higher elevation, noisy data). You present an estimate of uncertainty in your Figure 3 – it would be good to do the same in Figure 5.

Secondly, the link between your drainage/refreeze events and impact on ice velocity looks highly subjective and are difficult to interpet and verify from Figure 5 as you do not separate these events by elevation (like you have done with ice velocity) – consider improving this figure.

You also refer to an ice velocity increase coinciding with a period of refreezing in July 2019 (Line 391). How do these two mechanisms work?

**Minor comments:**

***Units:*** You interchange your use of units, from $km^2$ to $m^2$. For example, when defining small meltwater features you use $km^2$ (e.g., 0.0495 $km^2$), however when presenting your results, you refer to your meltwater features in $m^2$. Please choose one unit for consistency and comparison of results.

***Catchment reference:*** You refer to the Russell/Leverett catchment throughout the manuscript and provide its outline in Figure 1. Is this catchment delineated yourself (e.g., via flow routing) or is this an already pre-defined catchment? If the former, please provide a method as to how this catchment was delineated and what datasets were used. If the latter, please cite the appropriate data source.

**Specific comments:**

Line 44 – *'SGLs generally form in early summer enlarge in area and depth between spring and summer as they accumulate water...'.* This sentence does not fully make sense. Maybe add an *'and'* before enlarge or *',enlarging in area...'*.

Line 60 – could include additional references to remote sensing studies here (Lu et al., 2021; Turton et al., 2021; Rawlins et al., 2023; Zhang et al., 2023).

Lu, Y., Yang, K., Lu, X., Li, Y., Gao, S., Mao, W. and Li, M.: Response of supraglacial rivers and lakes to ice flow and surface melt on the Northeast Greenland ice sheet during the 2017 melt season, J. Hydrol. 602, 126750, 2021. https://doi.org/10.1016/j.jhydrol.2021.126750

Turton, J. V., Hochreuther, P., Reimann, N., and Blau, M. T.: The distribution and evolution of supraglacial lakes on 79° N Glacier (north-eastern Greenland) and interannual climatic controls, The Cryosphere, 15, 3877–3896, https://doi.org/10.5194/tc-15-3877-2021, 2021. https://doi.org/10.5194/tc-15-3877-2021

Line 73 – *'...the drainage of meltwater features was not considered.'* Drainage how? I am assuming drainage subglacially, but could be more explicit here as some of the papers cited assess how meltwater moves or is 'supraglacially-drained' across the surface over a single/multiple melt seasons.

Line 95 *(Figure 1)* - Your figure shows maximum areal extent of meltwater features in 2018 and 2019. Is this from a particular date in the season or an amalgamation of your features from separate mapped dates across the season into one map?

Line 102 – This study area is well known for its prevalent surface hydrology features including lakes, rivers, and moulins. Could you additionally provide an upper elevation estimate from previous studies (including refs)?

Line 143 – the blue and green filter you use. What is this? A band combination? Some clarity would help.

Line 146 – Include the citation and subsequent reference for McFeeters (1996) – paper for the traditional NDWI index (using green and NIR bands).

McFEETERS, S. K. (1996) 'The use of the Normalized Difference Water Index (NDWI) in the delineation of open water features', International Journal of Remote Sensing, 17(7), pp. 1425–1432. https://doi.org/10.1080/01431169608948714.

Line 171 – how narrow were the channels that were manually added? Are they still larger, primary rivers (as commented on earlier)?

Line 260 – would be helpful to give a clarifying statement as to the purpose of ice velocity data for the study to make this clearer.

Line 285 – You refer to both linear stream and river features. Maybe just state rivers (as per my previous comments).

Line 412 – Whilst I think it is relatively clear the contrast between surface meltwater characteristics and distribution (which would be helped by an additional figure) between years, I think I would refrain from saying there is a 'clear contrast' for drainage dynamics, as this looks to be subjective.

Table S1 – From the main body it was stated that 'Images with > 50% cloud cover were omitted' (Line 122). However, in Table S1, scene IDs have been included with cloud cover >50%. Do these scene IDs

need to be removed from the table? Or were these scene IDs (>50% cloud) used? If they were used, rectification in the main body is required.

**Technical corrections:**

Line 126 – Capitalise 'Level'

Line 131 – Missing bracket for citation

Line 133 – decide whether to capitalise (or not) the word 'bands'. Some inconsistencies.

Line 139 – you have already given the abbreviation for Normalised Difference Water Index (NDWI) on Line 89. You could therefore remove this on Line 139 if you wish (however, if retaining, please capitalise 'Normalised Difference Water Index' for consistency).

Line 217 – replace 'ocean' with lake or SGL

Line 296 – At the end of the sentence, either remove 'at' or the brackets.

In places, you have a citation and Figure number in separate brackets in immediate succession. Combine these using a semi-colon (Corr et al., 2022; Fig. 2).

---

## Author Comment (AC1)

Dear Dr Tedstone,

Thank you for taking the time to provide a review of our paper. Whilst we will provide a full, detailed, response to all of your comments in due course, we thought it would be helpful to provide an initial response to your main points here.

Your comments are copied below in normal text, and our response is given in blue.

**Summary of study**

This study maps the seasonal evolution of supraglacial meltwater features in a surface drainage catchment on the western margin of the Greenland ice sheet during two melt seasons by digitising the features from multispectral satellite scenes. Next, it seeks to attribute meltwater disappearance to either drainage or refreezing. Finally, it examines the case for links between surface meltwater drainage and basal sliding.

For clarity, the main aim of our study is to *compare* the distribution and morphology of supraglacial meltwater features in the Russell/Leverett glacier catchment in the low melt year of 2018, to the extreme high melt year of 2019. This is important as the frequency of high melt years like 2019 will likely increase in the future due to rising air temperatures. As such, we undertake detailed mapping of meltwater features to elucidate differences in their meltwater characteristics and dynamics under expected climatological conditions now (i.e. 2018) and in the future (i.e. 2019). In particular, we emphasise here that we are the first to map lakes < 0.0495 km$^2$ and - to the best of our knowledge - the first to include small lakes, large lakes, rivers/streams AND slush in a single inventory. In our revised paper, we will ensure that our study's key, unique, objectives are highlighted more clearly.

The purported main findings are that (i) surface meltwater features evolve depending on surface melt, draining from higher elevations in warmer summers and (ii) the drainage of these features, even very small ones, can lead to transient ice velocity responses, which may therefore be important to the future stability of ice flow in response to enhanced melting.

Our main findings are that in the higher melt year (2019): 1) surface meltwater features form and drain at higher elevations, 2) that small lake formation and drainage is more prevalent and 3) that slush is more widespread. The implication of this is that these patterns are likely to become the norm in future, higher melt years. In order to investigate the potential impact of this, we introduce ancillary data (i.e. ITS_LIVE velocity and PROMICE proglacial discharge) which suggest that these surface melt and drainage patterns do influence ice velocity and proglacial discharge in a different way in high melt years (our future analogue) than in low melt years.

**Originality/novelty**

The mapping of supraglacial meltwater features here uses established techniques which have been previously applied to a range of moderate and fine resolution imagery (non-exhaustively: Williamson et al., 2018; Smith et al., 2015). In this respect, I did not really learn anything new about the basic evolving pattern of supraglacial drainage in this area of the ice sheet that has not already been evidenced elsewhere.

Our study does not seek to develop new knowledge about the 'basic evolving pattern of supraglacial lake drainage' which as you say, is well documented elsewhere. Rather we seek to

discover differences in this pattern of supraglacial meltwater and drainage between high and low melt years, and we do find substantial differences, as explained in our response above.

During both melt years, however, we do make the novel finding that small lakes - disregarded in previous work (e.g. Williamson et al., 2018, Miles et al., 2017), because they are thought to be too small to drain - are able to drain rapidly and thus act as conduits for surface to bed meltwater transport in previously unexpected areas. This is important, as these conduits (aka moulins) will likely stay open for the remainder of the melt season.

**Scientific quality/rigour**

*(i) Ice velocities*

The authors employ the ITS_LIVE velocity dataset, which is derived from feature tracking of optical satellite imagery. The details provided in this study's methods are insufficient. For instance: what are the uncertainties/errors? Was any filtering (as opposed to smoothing) of the ITS_LIVE data carried out? 10 days is a very short baseline considering the expected ice displacement over this period in a "slow" land-terminating catchment and also in relation to the imagery pixel size, so is surely prone to high uncertainties.

Concretely, without these details I am especially concerned by the analysis of retrievals above ~1,200m asl. Among the several studies which have looked in detail at feature-tracked velocity retrievals in this area (e.g. Tedstone et al., 2015; Williams et al., 2020; Halas et al., 2022), it is clear that retrievals become very sparse to non-existent above 1,200-1,400 m asl, even when employing higher-resolution Sentinel-2 data, owing to a lack of features which can be reliably tracked. Of course, these studies examined annual net ice flow, not intra-annual flow, so are not directly comparable, but provide a conservative sense of the coverage of feature-tracked retrievals in this area of the ice sheet (or possibly even optimistic given the way in which they mosaic several acquisitions together)

In this light I was very surprised to see "unvalidated" use of ITS_LIVE velocities all the way up to the 2,000 m contour. Using the ITS_LIVE Binder (https://mybinder.org/v2/gh/nasa-jpl/its_live/main?urlpath=lab/tree/notebooks), I took a look at a single point of these data through time at roughly this elevation:

It is clear that at high elevations the short-baseline retrievals are exceptionally noisy. Previous GNSS observations in this sector (e.g. Sole et al., 2013; Doyle et al., 2014) have shown that ice speeds above 1,500 m asl are less than c. 100 m/yr and show maximum daily velocities of up to max. 300 m/yr, generally occurring over periods less than 10 days (based on Sole Fig. 2). So, there are lots of velocity retrievals above which are simply not supportable by reference to previous ground observations.

To be satisfied that the analysis presented in the present study is appropriate, I would need to see evidence of: (a) a robust filtering approach to treat the abundant outliers (not just a boxcar moving window as currently used to smooth the data); (b) ideally, examination of the underlying signal-to- noise ratio (i.e. does the ITS_LIVE algorithm even support these velocities?); (c) an error budget/uncertainty analysis. (d) At the highest elevations, I suggest to go further, interrogating the velocity fields with reference to the underlying imagery, as I suspect that spurious cross-correlations are being identified associated with ephemeral slush fields, which are much less 'stable' than the ice- incised supraglacial channels found at lower elevations.

Put simply, at the moment I do not believe the velocity analysis for elevations higher than ~1,200 m/yr.

We appreciate your concerns about our use of the ITS_LIVE velocity data, however we think you may have inferred greater weight imposed upon its inclusion than intended in the paper. We will ensure this is clarified in our revised paper.

To clarify here in this letter, our aim is to use this as ancillary data to support our analysis, as opposed to a primary dataset upon which we construct our original arguments. This is in a similar manner to the use of ITS_LIVE data in numerous other, published, studies (e.g. Otto et al., 2022; Wang and Sugiyama, 2024, Arthur et al., 2021; Boxall et al., 2022). We note that the data only appears in one of the five figures in the paper, and we are careful to use language such as 'appears to perturb ice velocity' within the text, which is in recognition of its uncertainties.

We do perform some filtering: we only include data within the 1st and 99th percentile in order to remove the outliers you mention, and we can certainly include the uncertainty estimates provided to us by ITS_LIVE. Both of these will be elucidated in the revised manuscript.

We are also happy to test the sensitivity of the signals we see to our choice of baseline and filtering parameters, and to include details of this testing in a revised supplement to the main paper. However, an in-depth independent validation of the dataset beyond this is well outside the scope of this study.

**(ii) Surface hydrology partitioning**

Like with the ice velocities, my concerns particularly relate to higher elevations of the catchment. The study uses air temperatures to apportion the disappearance of surface meltwater into either drainage or refreezing. When meltwater disappears between two successive satellite acquisitions, if air temperatures were positive it is assumed to drain, whereas if they were negative it is assumed to refreeze. This is almost certainly overly simplistic. First, for instance, going all the way back to Holmes (1955), there is evidence that meltwater can continue to flow in open channels for up to two weeks after the end of surface melting. There is therefore a substantial lag between the onset of negative air temperatures and the freeze-up of the surface.

We agree that using air temperature to partition between drainage and refreezing is rather simplistic, especially with respect to the larger lakes which possess sufficient thermal inertia to resist refreezing for some time. However, manual inspection of the imagery suggests that it is effective. In the revised manuscript we are happy to validate the partitioning by cross referencing the 'refreezing' lakes with SAR imagery, which will enable us to confirm where liquid water persists below the surface and has not drained (e.g. Miles et al., 2017; Benedek et al., 2021; Dunmire et al., 2021).

Second, I suspect that there is some antialiasing of the evolution of surface drainage features with 'drainage'. This particularly concerns slush fields, which can either collapse/incise into more spatially discrete, efficient supraglacial channels – thereby presumably allowing water to be evacuated more quickly – or never evolve as far as incision, instead allowing water to continue to flow through the matrix. In this case, just because the water disappears from view, does not mean that all that water has refrozen. Instead, it may be discharging via the subsurface, as shown by Clerx et al. (2022), in a water table that is below the height of the snow surface.

Regarding slush in 2019, we agree that this can drain via the two mechanisms you mention. We do not see any consolidation into discrete channels in our data so 'drainage' is likely to be downstream discharge through the water table. We do discuss the downstream discharge of slush on lines 485-488 of the paper, but we accept that this could be made clearer with respect to the physical mechanism, and we will edit this accordingly in the revised manuscript.

We think that our characterisation of 22% of slush area as refreezing using the comparison with RACMO is reasonable given that a) it occurs high up in the percolation zone where overland flow is limited and thus if melting stops then water flow stops, b) water held within an ice matrix (i.e. slush) is likely to refreeze much more readily than water within a large open-water body and c) the timing of this is associated with the onset of refreezing of the entire catchment at the end of the melt season, which begins at these high elevations.

Secondarily, I am concerned about the suitability of the water depth retrieval algorithm to drainage at higher elevations. These algorithms were developed on the basis of a solid ice substrate, which is often not the case above the ELA. In particular, slush fields are composed of a porous water-filled snow matrix perched on top of an ice slab formed of refrozen meltwater (see e.g. Clerx et al., 2022). Thus, neither are they spectrally similar to supraglacial ice-incised channels, nor do their depths correspond to an entirely liquid column. This makes the water volume retrievals from these features problematic.

We agree that the radiative transfer approach to measuring lake depth is suboptimal in terms of measuring *absolute* lake depth, and indeed many of our team were involved in a recent study examining just this (Melling et al., 2024). We think it is reasonable however to assume that *relative changes* to volume between successive scenes (i.e. required by the FASTER algorithm to identify drainage) are relatively robust. Regardless of this, many of the patterns we see are obvious in the lake area data alone (e.g. in figure 5), so we are happy to remove reference to the volume data in the revised manuscript where we use it in the absolute sense and where it is redundant with respect to associated area data.

I note that some data are indicated as 'on request'. According to TC submission requirements, this is not acceptable.

We have uploaded our dataset here: https://zenodo.org/doi/10.5281/zenodo.10949984. This dataset will be referenced in the revised manuscript.

References

Arthur, J. F., Stokes, C. R., Jamieson S, S. R., Miles, B. W. J, Carr, J. R., Leeson, A. A.: The triggers of the disaggregation of Voyeykov Ice Shelf (2007), Wilkes Land, East Antarctica, and its subsequent evolution, *Journal of Glaciology*, 67(265):933-951, doi:10.1017/jog.2021.45, 2021.

Benedek, C. L. and Willis, I. C.: Winter drainage of surface lakes on the Greenland Ice Sheet from Sentinel-1 SAR imagery, The Cryosphere, 15, 1587–1606, https://doi.org/10.5194/tc-15-1587-2021, 2021.

Boxall, K., Christie, F. D. W., Willis, I. C., Wuite, J., and Nagler, T.: Seasonal land-ice-flow variability in the Antarctic Peninsula, The Cryosphere, 16, 3907–3932, https://doi.org/10.5194/tc-16-3907-2022, 2022.

Dunmire, D., Banwell, A. F., Wever, N., Lenaerts, J. T. M., and Datta, R. T.: Contrasting regional variability of buried meltwater extent over 2 years across the Greenland Ice Sheet, The Cryosphere, 15, 2983–3005, https://doi.org/10. 5194/tc-15-2983-2021, 2021.

Melling, L., Leeson, A., McMillan, M., Maddalena, J., Bowling, J., Glen, E., Sandberg Sørensen, L., Winstrup, M., and Lørup Arildsen, R.: Evaluation of satellite methods for estimating supraglacial lake depth in southwest Greenland, The Cryosphere, 18, 543–558, https://doi.org/10.5194/tc-18-543-2024, 2024.

Miles K. E., Willis I. C., Benedek, C. L., Williamson, A. G., and Tedesco, M.: Toward Monitoring Surface and Subsurface Lakes on the Greenland Ice Sheet Using Sentinel-1 SAR and Landsat-8 OLI Imagery, Front. Earth Sci. 5:58, https://doi.org/10.3389/feart.2017.00058, 2017.

Otto J, Holmes, F. A. and Kirchner, N.: Supraglacial lake expansion, intensified lake drainage frequency, and first observation of coupled lake drainage, during 1985–2020 at Ryder Glacier, Northern Greenland. Front. Earth Sci. 10:978137, doi: 10.3389/feart.2022.978137, 2022

Wang, Y., Sugiyama, S.: 'Supraglacial lake evolution on Tracy and Heilprin Glaciers in northwestern Greenland from 2014 to 2021', Remote Sensing of Environment, Volume 303, 2024, 114006, ISSN 0034-4257, https://doi.org/10.1016/j.rse.2024.114006, 2024.

Williamson, A. G., Banwell, A. F., Willis, I. C., and Arnold, N. S.: Dual-satellite (Sentinel-2 and Landsat 8) remote sensing of supraglacial lakes in Greenland, The Cryosphere, 12, 3045–3065, https://doi.org/10.5194/tc-12-3045-2018, 2018.

---

## Author Comment (AC2)

**Author response to Reviewer 1**

Dear Dr Tedstone,

Thank you for taking the time to provide a detailed review of our paper. Our response below builds on our initial, brief response submitted on 15 Apr 2024. Your comments are copied below in black text, and our responses are in blue.

**Summary of study**

This study maps the seasonal evolution of supraglacial meltwater features in a surface drainage catchment on the western margin of the Greenland ice sheet during two melt seasons by digitising the features from multispectral satellite scenes. Next, it seeks to attribute meltwater disappearance to either drainage or refreezing. Finally, it examines the case for links between surface meltwater drainage and basal sliding.

For clarity, the main aim of our study is to *compare* the distribution, morphology and evolution of supraglacial meltwater features in the Russell/Leverett glacier catchment in the low melt year of 2018, to the extreme high melt year of 2019. This is important as the frequency of high melt years like 2019 will likely increase in the future due to rising air temperatures.

In particular, we emphasise here that we, to the best of our knowledge, are the first to include small lakes (< 0.0495 km²), large lakes (> 0.0495 km²), AND slush in a single inventory. Now in our revised paper, following Reviewer 2's suggestion, we also map supraglacial channels to produce an even more inclusive inventory of surface meltwater evolution over the two melt seasons (for further detail, see our comments on page 7 of our response letter to R2).

In our revised paper, we will ensure that our study's key, unique, objectives are highlighted more clearly. For example, we will make our introduction clearer to include a revised paragraph along the lines of the following: '*Our unique dataset includes small (i.e., <0 .0495 km²) and shallow meltwater features (including slush), which are important, but have to date been overlooked in previous mapping studies, as well as the previously more commonly included, large lakes (i.e., > 0.0495 km²) and channels (i.e., all linear meltwater features > 1000 m long). We compare the seasonal evolution of supraglacial meltwater features in low and high melt seasons, with a focus on meltwater feature characteristics and drainage dynamics, and partitioning features into those that drain and refreeze. We then assess the potential implications of differences between the high and low melt year by introducing ancillary data (i.e. ITS_LIVE velocity and PROMICE proglacial discharge), enabling us to gain an insight into how a warming climate - where the high melt year becomes the norm - might impact meltwater throughput and ice flow, as a result of changes to surface meltwater characteristics and lake drainage dynamics.*'

The purported main findings are that (i) surface meltwater features evolve depending on surface melt, draining from higher elevations in warmer summers and (ii) the drainage of these features, even very small ones, can lead to transient ice velocity responses, which may therefore be important to the future stability of ice flow in response to enhanced melting.

We agree with you that these are *some* of our findings. However, more importantly, our findings relate to differences between the two melt years, including that in the higher melt year (2019): 1) surface meltwater features form and drain at higher elevations, 2) small lake formation and drainage is more prevalent and 3) slush is more widespread. The implication of this is that these patterns are likely to become the norm in future, higher melt years. Interestingly, we also have the unique finding that meltwater features in 2018 (the low melt year) tend to be deeper and more voluminous than in 2019 (the high melt year), as shown in the new figure below, which we will add to the Supplement of the revised paper. This is a finding that we will discuss in our revised results/discussion.

[Figure]

*NEW supplementary figure: Box plots showing (a) maximum volume, (b) maximum area, (c) maximum depth, (d) mean depth, (e) ice thickness and (f) elevation of surface lakes in 2018 (N = 1011) and 2019 (N = 1495). Red is the median and the edges of the box are the 25th and 75th percentiles ($q_1$ and $q_3$, respectively).*

**Major comments**

My overall view is that this study is weak. The basic methodology is not novel: mapping of supraglacial meltwater features is well-established in the literature, indeed as highlighted by the references in the study. This would not be a problem if the rest of the analysis made solid contributions on top of this approach, but I did not find this to be the case. I am unconvinced by the partitioning between drainage and refreezing, while the links with catchment wide ice velocity are in principle novel but appear to suffer from some major methodological flaws which I fear are unfixable. Please find more reasoning for my conclusions below. I am sorry that I cannot be more positive about the manuscript at this time. I am open to discussion if the authors feel I have mis-interpreted aspects of the study.

It is correct that we base our methodology on previously established and validated methods used successfully in previous mapping studies, and indeed we reference these studies throughout our manuscript. It should be noted, however, that we do adapt the well-established NDWI method to add small and shallow meltwater features to our dataset. In our revised manuscript, we will also adjust the threshold to delineate slush in order to detect even shallower slush features than previously Additionally, in our revised manuscript, we will expand our analysis to also include delineation of surface channels (defined to include both rivers and streams > 1000 m long), following Yang et al. (2016). Please see our separate response letter to Reviewer 2 for more details about our additional channel analysis and results, as well as our revised slush delineation analysis and results (pages 3 -7) of the R2 response letter).

**Originality/novelty**

The mapping of supraglacial meltwater features here uses established techniques which have been previously applied to a range of moderate and fine resolution imagery (non-exhaustively: Williamson et al., 2018; Smith et al., 2015). In this respect, I did not really learn anything new about the basic evolving pattern of supraglacial drainage in this area of the ice sheet that has not already been evidenced elsewhere.

Our study does not seek to develop new knowledge about the 'basic evolving pattern of supraglacial lake drainage' which as you say, is well documented elsewhere. Rather, as also mentioned earlier in this letter, we seek to discover differences in this pattern of supraglacial meltwater and drainage between high and low melt years, and we do find substantial differences, as explained in our response above. In our revised paper, we will ensure to make this overriding objective of our study clearer. Additionally, during both melt years, we also make the novel finding that small lakes - disregarded in previous work (e.g. Williamson et al., 2018, Miles et al., 2017) because they are thought to be too small to drain - are able to drain rapidly and thus form moulins that act as conduits for surface to bed meltwater transport in previously unexpected areas. This is especially important as these moulins will likely stay open, and hence will act as surface to bed connections, for the remainder of the melt season (e.g. Banwell et al 2016).

As I indicated above, the link to ice velocity has the potential to be novel, as does the attribution to meltwater disappearance between refreezing and drainage. However, I have major concerns about the quality of these two parts of the analysis.

**Scientific quality/rigour**

Here I concentrate on the two areas of analysis which I found most problematic.

*(i) Ice velocities*

The authors employ the ITS_LIVE velocity dataset, which is derived from feature tracking of optical satellite imagery. The details provided in this study's methods are insufficient. For instance: what are the uncertainties/errors? Was any filtering (as opposed to smoothing) of the ITS_LIVE data carried out? 10 days is a very short baseline considering the expected ice displacement over this period in a "slow" land-terminating catchment and also in relation to the imagery pixel size, so is surely prone to high uncertainties.

Concretely, without these details I am especially concerned by the analysis of retrievals above ~1,200m asl. Among the several studies which have looked in detail at feature-tracked velocity retrievals in this area (e.g. Tedstone et al., 2015; Williams et al., 2020; Halas et al., 2022), it is clear that retrievals become very sparse to non-existent above 1,200-1,400 m asl, even when employing higher-resolution Sentinel-2 data, owing to a lack of features which can be reliably tracked. Of course, these studies examined annual net ice flow, not intra-annual flow, so are not directly comparable, but provide a conservative sense of the coverage of feature-tracked retrievals in this area of the ice sheet (or possibly even optimistic given the way in which they mosaic several acquisitions together)

In this light I was very surprised to see "unvalidated" use of ITS_LIVE velocities all the way up to the 2,000 m contour. Using the ITS_LIVE Binder (https://mybinder.org/v2/gh/nasa-jpl/its_live/main?urlpath=lab/tree/notebooks), I took a look at a single point of these data through time at roughly this elevation:

It is clear that at high elevations the short-baseline retrievals are exceptionally noisy. Previous GNSS observations in this sector (e.g. Sole et al., 2013; Doyle et al., 2014) have shown that ice speeds above 1,500 m asl are less than c. 100 m/yr and show maximum daily velocities of up to max. 300 m/yr, generally occurring over periods less than 10 days (based on Sole Fig. 2). So, there are lots of velocity retrievals above which are simply not supportable by reference to previous ground observations.

To be satisfied that the analysis presented in the present study is appropriate, I would need to see evidence of: (a) a robust filtering approach to treat the abundant outliers (not just a boxcar moving window as currently used to smooth the data); (b) ideally, examination of the underlying signal-to- noise ratio (i.e. does the ITS_LIVE algorithm even support these velocities?); (c) an error budget/uncertainty analysis. (d) At the highest elevations, I suggest to go further, interrogating the velocity fields with reference to the underlying imagery, as I

suspect that spurious cross-correlations are being identified associated with ephemeral slush fields, which are much less 'stable' than the ice- incised supraglacial channels found at lower elevations. Put simply, at the moment I do not believe the velocity analysis for elevations higher than ~1,200 m/yr.

We appreciate your concerns about our use of the ITS_LIVE velocity data, however we suggest that you may have inferred greater weight imposed upon its inclusion than intended in the paper. To clarify, our aim is to use these ancillary data to support our analysis, as opposed to a primary dataset upon which we construct our original arguments; we will ensure this is also clarified in our revised paper. Our aim and methodology is similar to the use of ITS_LIVE data in numerous previous, published, studies (e.g. Otto et al., 2022; Wang and Sugiyama, 2024, Arthur et al., 2021; Boxall et al., 2022). We note that the data only appears in one of the five figures in the paper, and we are careful to use language such as 'appears to perturb ice velocity' within the text, which is in recognition of its uncertainties.

We do perform some filtering: we only include data within the 1st and 99th percentile in order to remove the outliers you mention, and we include the uncertainty estimates provided to us by ITS_LIVE in our revised Figure 5. We will refer to our filtering method and addition of uncertainty estimates in the revised manuscript.

More specifically, in our revised manuscript, we will add the following text to the methods section:

*"In line with several other studies, we use ITS_LIVE data to support our analysis (e.g. Otto et al., 2022; Wang and Sugiyama, 2024, Arthur et al., 2021; Boxall et al., 2022)."*

*"We filter the data between the 1st and 99th percentile in order to remove the outliers and a three-day smoothing window was used to further reduce noise."*

*"Uncertainty measurements are provided by ITS_LIVE and are detailed in the ITS_LIVE Regional Glacier and Ice Sheet Surface Velocities documentation (Gardner et al., 2023). ITS_LIVE velocity data is retrieved through auto-RIFT feature tracking methods and we note that ice velocities retrieved using this method at high elevation regions (i.e., > 1200 m a.s.l) are subject to high uncertainties (e.g., Tedstone et al., 2015; Williams et al., 2020; Halas et al., 2022) due to the lack of trackable features here."*

Additionally, in response to your review comment, we have tested the sensitivity of the observed signals in the velocity data to our choice of baseline and filtering parameters and produced the new figure pasted below, which we will put in the Supplement. This figure compares a) filtering, b) image separation, and c) smoothing. However, an in-depth independent validation of the ITS_LIVE dataset beyond this is well outside the scope of this study.

[Figure]

*NEW supplementary figure: Sensitivity of 2019 ITS_LIVE velocity data to a) filtering with 0, 95th and 99th percentiles, b) image separation under 5-, 10- and 20-day time steps, and c) smoothing with 0-, 3- and 5-day moving windows. Data from 800 m a.s.l (left) and 2000 m a.s.l (right) are shown. Shaded regions indicate the uncertainty taken directly from the ITS_LIVE data product (Gardner et al., 2018; 2023).*

**(ii) Surface hydrology partitioning**

Like with the ice velocities, my concerns particularly relate to higher elevations of the catchment. The study uses air temperatures to apportion the disappearance of surface meltwater into either drainage or refreezing. When meltwater disappears between two successive satellite acquisitions, if air temperatures were positive it is assumed to drain, whereas if they were negative it is assumed to refreeze. This is almost certainly overly simplistic. First, for instance, going all the way back to Holmes (1955), there is evidence that meltwater can continue to flow in open channels for up to two weeks after the end of surface melting. There is therefore a substantial lag between the onset of negative air temperatures and the freeze-up of the surface.

We agree that our previous methods that we used for partitioning between drainage and refreezing, based on air temperature alone, were rather simplistic. In our revised manuscript, we will take two additional measures to improve our method of partitioning lake drainage vs. refreezing, and to account for associated uncertainty:

1) We now class a meltwater feature as refreezing not only based on air temperature, but also on the pattern of lake volume decline, following Selmes et al. (2013). In our original manuscript, lake drainage and refreezing were classified as such if > 20% of volume is lost over any time period within a melt season. And refreezing (rather than drainage) was previously assumed to occur if the air temperature was < 0C +/- 1C for at least three days prior (following *Wan et al., 2002 and Selmes et al. 2013)*. Now, in our new methods, we also

implement a > 72-hour time requirement, whereby refreezing is only assumed to occur if a lake takes at least 72 hours to lose 20% of its volume AND the air temperature stays below (or equal to) 0C. This additional measure reflects the fact that refreezing does not happen instantaneously after the onset of negative temperatures, as you state above.

2) We have also added an 'unknown' class to our lake behaviour partitioning (i.e. in addition to 'drain' and 'refreeze'), A lake is classified as 'unknown' if the pattern of area and volume decline is not in accordance with that of a draining or refreezing lake.

To provide some validation for our approach, we have cross referenced a subset of our refreezing lakes with two independent datasets of subsurface lakes acquired from SAR imagery in both years (Dunmire et al., 2021; Zheng et al., 2023), under the assumption that refrozen lakes essentially become subsurface lakes after refreezing. We find good agreement in the locations of refreezing lakes in our dataset when compared to the other two datasets (See NEW supplementary Figure on page 7 of this letter). We suggest that differences in the locations of these refrozen lakes are due to differing NDWI thresholds between the three datasets.

[Figure]

*NEW supplementary figure: Cross referencing a subset of refreezing lakes in this study with maps of subsurface lakes acquired from SAR imagery in both 2018 (left) and 2019 (right). Blue: dataset from Dunmire et al. (2021); green: dataset from Zheng et al. (2023); red: Data from this study. Coordinates of lakes in WGS 1984 UTM Zone 22N: a) 47.3861245°W 66.9952765°N, b) 48.4755826°W 67.0147779°N, c) 47.4456300°W 66.8319809°N, d) 48.2075289°W 67.1934572°N e) 47.3873357°W 66.9953815°N, f) 48.5521200°W 67.1620352°N.*

Using the optical satellite record, we have also manually identified ice formation on large lake surfaces that we classify as refreezing (e.g. image below). For these lakes, we also

manually check that surrounding meltwater features also appear to decrease in area, which points towards a common forcing mechanism (most likely refreezing).

[Figure]

*Example of a refreezing lake (for the purposes of this letter only) identified in a Sentinel-2 image from 17/08/2019. Note the formation of an ice lid and the formation of thin ice on the surface. Lake situated at 47.3873357°W 66.9953815°N (WGS 1984 UTM Zone 22N).*

In general, we consider our approach for identifying lakes that refreeze to be conservative relative to other studies. For example, as indicated by our new statistics (new Table 1 below as well as in our revised Figure 4 below), we calculate that 13% and 4% of lakes refreeze by the end of the 2018 and 2019 melt season, respectively. In contrast, Selmes et al. (2013) determined that on average, over multiple melt seasons, ~46 % of lakes froze. We will also state this point in the revised paper.

*REVISED Table 1: Meltwater statistics of draining and refreezing lakes as well as those with unknown behaviour in 2018 and 2019, where draining and refreezing features were identified using the FASTER algorithm (Williamson et al., 2018a) and subsequently partitioned using RACMO 2 m air temperature data (Noël et al., 2019), with a threshold of < 0 °C identified as refreezing. DOY is the 'day of year' in 2018 and 2019. DOY sampling is calculated by averaging the start drainage DOY and the end drainage DOY. Percentage values are proportions of the sum of the total meltwater areas or volumes for each melt season.*

|  | 2018 | | | 2019 | | |
| --- | --- | --- | --- | --- | --- | --- |
| **Statistic** | *Drainage* | *Refreeze* | *Unknown* | *Drainage* | *Refreeze* | *Unknown* |
| Frequency (n) | 432 | 129 | 450 | 650 | 59 | 786 |
| n (%) | 43 | 13 | 44 | 44 | 4 | 50 |
| Total Volume (km³) | 0.54 | 0.065 | 0.0035 | 0.50 | 0.12 | 0.0015 |
| Total Volume (%) | 89 | 11 | < 1 | 80 | 19 | 1 |
| Mean Volume (km³) | 1.3e-3 | 5.0e-4 | 7.9 e-6 | 7.7e-4 | 2.8e-4 | 1.8e-6 |
| Total Area (km²) | 59 | 12 | 3.5 | 99 | 23 | 3.5 |
| Total Area (%) | 80 | 16 | 4 | 79 | 19 | 2 |
| Mean Area (km²) | 0.14 | 0.091 | 0.0079 | 0.15 | 0.40 | 0.0044 |
| Mean Depth (m) | 1.7 | 1.4 | 0.73 | 1.4 | 1.8 | 0.53 |
| Mean event DOY | 195 | 200 | n/a | 159 | 166 | n/a |
| Mean DOY sampling (± days) | 6 | 6 | n/a | 2 | 5 | n/a |

[Figure]

*REVISED Figure 4: Drainage and refreezing within the Russell/Leverett glacier catchment in 2018 (left) and 2019 (right). (a) and (b) depict the timing of lake drainage (square) and lake refreezing (triangle) events in 2018 and 2019, respectively. Lakes of unknown drainage/refreezing behaviour are represented as small grey circles. (c) and (d) depict small (<0.0495 km²; black) and large (>0.0495 km²; white) lake drainage (square) and lake refreezing (triangle) events in 2018 and 2019, respectively. Light to dark purple gradient represents ice thickness in metres. Lakes of unknown drainage/refreezing behaviour are not shown in panels c and d.*

Second, I suspect that there is some antialiasing of the evolution of surface drainage features with 'drainage'. This particularly concerns slush fields, which can either collapse/incise into more spatially discrete, efficient supraglacial channels – thereby presumably allowing water to be evacuated more quickly – or never evolve as far as incision, instead allowing water to continue to flow through the matrix. In this case, just because the water disappears from view, does not mean that all that water has refrozen. Instead, it may be discharging via the sub-surface, as shown by Clerx et al. (2022), in a water table that is below the height of the snow surface.

We agree with your concern, so we now only focus on partitioning the drainage and refreezing of lakes; not slush. Although we still analyse the seasonal evolution of lake, channel and slush features, we remove slush features from our drainage/refreezing dataset to avoid any confusion between the evolution of surface features with drainage.

Secondarily, I am concerned about the suitability of the water depth retrieval algorithm to drainage at higher elevations. These algorithms were developed on the basis of a solid ice substrate, which is often not the case above the ELA. In particular, slush fields are composed of a porous water-filled snow matrix perched on top of an ice slab formed of refrozen meltwater (see e.g. Clerx et al., 2022). Thus, neither are they spectrally similar to supraglacial ice-incised channels, nor do their depths correspond to an entirely liquid column. This makes the water volume retrievals from these features problematic.

We agree that there are uncertainties when applying the radiative transfer model to slush, for the reasons you state. Therefore, as we also stated above, we no longer use the radiative transfer model to calculate the depth of slush (and nor channels); we only apply it to lakes.

Additionally, we appreciate that even though we now do not apply the radiative transfer model to slush, there will be some uncertainty when applying the radiative transfer model to lakes above the ELA, especially as these lakes are surrounded by slush. For example, many of our team were involved in a recent study examining the accuracy of this algorithm (Melling et al., 2024), which we will comment on in our revised paper. We also note that we do include the uncertainty associated with our lake volume calculations, as shown in our revised Figure 3, pasted below.

[Figure]

*REVISED Figure 3: Time series of total areas of lakes (dark blue), channels (green), slush (light blue) and all meltwater features (grey) in (a) 2018 and (b) 2019 from L8 and S2 imagery. Area error bars represent uncertainty of automatically delineated features compared to a manually delineated dataset. Lake volume is given in red error along with an estimate of uncertainty determined by comparing lake depth to Melling et al. (2023). Also shown is cloud cover percentage (black bars), RACMO 2 m air temperature anomaly (light green line) from the 1958 - 2019 catchment average with the spatial standard deviation (light green shading), and RACMO total daily melt (mm w.e.; light blue line) with the spatial standard deviation (light blue shading). Note that the y-axis ranges are different for the channel and slush areas between (a) and (b).*

Given my perspective above, in my view this study is not able to make impactful insights into the fate of supraglacial meltwater on the Greenland Ice Sheet.

We hope that our detailed responses above help you to understand how our study, and in particular our revised paper, does give impactful insights into the fate of supraglacial meltwater on the Greenland Ice Sheet when a high and low melt season are compared.

**Presentation quality**

This is overall reasonable. The manuscript is mostly clearly written. I suspect it could be made more concise in places (i.e. length reduced). The figures are basically fine, apart from Figure 5, which presents ice velocities by elevation band but does not also segment the drainage/refreeze events/area/volumes by elevation band. This makes it very difficult if not impossible to independently verify the proposed links between surface hydrology "events" and ice velocity/basal sliding.

We have revised our previous Figure 5 as per your recommendations, and those of Reviewer 2. Our revised Figure 5 is pasted below.

[Figure]

*REVISED Figure 5: Time series of lake drainage and refreeze within the Russell/Leverett catchment in a) 2018 and b) 2019. From top to bottom: daily frequency of lake drainage/refreeze events (i.e., the number lakes that drained or refroze); total daily lake area loss; total daily volume loss; mean ice velocity at 800 m a.s.l (red), 1200 m a.s.l (orange), 1600 m a.s.l (blue) and 2000 m a.s.l (purple). Shading indicates the uncertainty taken directly from the ITS_LIVE data product (Gardner et al., 2018; 2023). Vertical grey shaded columns depict, and velocity perturbations discussed in the text (vertical grey shaded columns, labelled i, ii, iii, and iv); Daily values of meltwater discharge through the Watson River (black line) and associated uncertainty (grey shading). Note that the x-axis date ranges are different for each year, constrained by the first and last meltwater feature drainage events in each melt season. Also note that the y-axis for the velocity plots differ between elevation bands.*

I note that some data are indicated as 'on request'. According to TC submission requirements, this is not acceptable.

We have now uploaded our updated dataset here: https://zenodo.org/doi/10.5281/zenodo.11645884, which will be referenced in the revised manuscript.

**Minor comments**

In light of my major comments, I have only a small number of minor comments at this point. Clarification of terminology: particularly in the methods, in general 'surface meltwater features' is employed, but sometimes 'lakes' or 'slush fields' are used instead. This is particularly the case in sect. 2.6, which initially claims to track 'meltwater features' but then uses this term and 'lakes' interchangeably. See also L175, 'accuracy of lake area estimates' concerning delineations, was this actually only for lakes (and not also channels etc), and if so, why?

We will be clearer throughout our revised manuscript when referring to all meltwater features (now including lakes, channels and slush), and also when referring to separate lake, channel and slush features. We have undertaken an accuracy assessment of lake, channel and slush areas, indicated in the uncertainty calculations in our revised Figure 3 (page 10). We did this by comparing automatically delineated lake, channel and slush features with a fully manually delineated dataset.

L185: surface gradients of meltwater features were retrieved from ArcticDEM. Presumably this is highly sensitive to whether those features were water-filled at the time of data acquisition for the DEM? More details are needed to assess whether this is a valid approach.

We agree that the error in the DEM may be too high for this method to be a valid approach for calculating slopes of small surface meltwater features, so we have removed all reference to surface gradients of these features from the text.

Sect. 2.7, use of RACMO: Given the high quality of in-situ AWS measurements on the K-Transect, it is valid to consider/state the performance of RACMO along this transect. Referencing should be sufficient.

We will add: '*RACMO performs well compared to automatic weather station data acquired in the same region, i.e. along the K-Transect (Noël et al., 2018).*'

L389 and around: references to panel a of Fig. 5, but this is to drainage only, without also referencing the velocities panel. Please improve.

We will edit this to refer to both drainage and velocity.

L540-1: 'perturb ice velocity at lower elevations…this is unexpected'. I disagree. Other studies, for example, Doyle et al. (2014) and Ryan et al. (2024), show that transient velocity variations occur whenever the subglacial drainage system's capacity is overwhelmed by the rate of meltwater supply . Rather than considering "drainage efficiency" to be an absolute quantity, consider it instead relative to antecedent and event melt supply.

Thank you for pointing this out. We agree with you, and will take this comment into consideration in our revised manuscript.

References not in the original paper (which we will add to our revised paper)

Arthur, J. F., Stokes, C. R., Jamieson S, S. R., Miles, B. W. J, Carr, J. R., Leeson, A. A.: The triggers of the disaggregation of Voyeykov Ice Shelf (2007), Wilkes Land, East Antarctica, and its subsequent evolution, *Journal of Glaciology*, 67(265):933-951, doi:10.1017/jog.2021.45, 2021.

Benedek, C. L. and Willis, I. C.: Winter drainage of surface lakes on the Greenland Ice Sheet from Sentinel-1 SAR imagery, The Cryosphere, 15, 1587–1606, https://doi.org/10.5194/tc-15-1587-2021, 2021.

Boxall, K., Christie, F. D. W., Willis, I. C., Wuite, J., and Nagler, T.: Seasonal land-ice-flow variability in the Antarctic Peninsula, The Cryosphere, 16, 3907–3932, https://doi.org/10.5194/tc-16-3907-2022, 2022.

Otto J, Holmes, F. A. and Kirchner, N.: Supraglacial lake expansion, intensified lake drainage frequency, and first observation of coupled lake drainage, during 1985–2020 at Ryder Glacier, Northern Greenland. Front. Earth Sci. 10:978137, doi: 10.3389/feart.2022.978137, 2022

Wang, Y., Sugiyama, S.: 'Supraglacial lake evolution on Tracy and Heilprin Glaciers in northwestern Greenland from 2014 to 2021', Remote Sensing of Environment, Volume 303, 2024, 114006, ISSN 0034-4257, https://doi.org/10.1016/j.rse.2024.114006, 2024.

Glen, E. (2024). Dataset for: A comparison of supraglacial meltwater features throughout contrasting melt seasons: Southwest Greenland [Data set]. Zenodo. https://doi.org/10.5281/zenodo.11645884

---

## Author Comment (AC3)

**Author response to Reviewer 2**

We thank this reviewer for their very useful and carefully considered comments that will help to improve our paper. Their comments are repeated here in black text and our replies are in blue below.

**General comments:**

This manuscript investigates the seasonal evolution of supraglacial meltwater features in the Russell/Leverett glacier catchment, SW Greenland, with a focus on drainage distribution and characteristics from a low (2018) and high (2019) melt year and assess its implications, including ice velocity and potential drainage response in future warmer years. Of particular interest is the attempt by the authors to include small (i.e., <0.0495 km$^2$) and shallow meltwater features (e.g., slush), which can be overlooked in mapping studies, however important to consider. The authors use a pre-existing method (Corr et al., 2022) for all supraglacial meltwater feature extraction and then seek to partition these features into those that drain and refreeze, with links to ice velocity events.

The main findings are that (i) surface meltwater feature characteristics and distribution differ between a high (2019) vs low (2018) melt season, with meltwater features developing earlier (May) and occurring further inland (2000 m a.s.l) in the high melt season (2019); (ii); small meltwater features (<0.0495 km$^2$), predominately small SGLs, are important features of the system with their drainage prevalent at lower elevations and; (iii) the drainage of features, including those that are small, can generate an ice velocity response, with inference that a sustained speed-up in ice flow may occur at this catchment in the future.

We are pleased to see that this reviewer has understood the main findings of our manuscript and also that they appreciate our approach of including small meltwater lakes and understand the importance of these features.

Whilst I appreciate the effort put into the development of this study and manuscript, my view is that this study is not particularly novel in terms of the methods used (well-versed in literature) or the location of the study area in SW Greenland, which has been well documented and the focus of many supraglacial hydrology studies over recent years. Additionally, whilst you present the importance of small meltwater features (<0.0495 km$^2$), particularly small SGLs, and appreciate why they were included in terms of your findings (as agree, they are important!), I struggle with your use of the term '*all meltwater features*'. From Figure 1, it seems your river and stream network is lacking (even with the resolution of imagery used), with only larger portions of rivers extracted, often associated with either inputting or outputting an SGL (interpreted from Figure 1). Therefore I think your use of the term '*all meltwater features*' insinuates complete supraglacial drainage network maps and analysis (including smaller, shallower features such as the river and stream network) across the Russell/Leverett catchment, which is misleading as believe this is a basic supraglacial network at best. I therefore think there needs to be increased clarity of this or further mapping to be undertaken.

We agree that the specific methods we use are not particularly novel, however developing novel methods was not the aim of our study. Having said that, we note that we do adapt methods by the addition of extensive manual enhancement in order to accurately delineate small meltwater features, and in our revised paper, we also adjust NDWI thresholds to better distinguish shallow meltwater.

We agree that our original approach was limited in its ability to delineate channels. To address this, we will add delineated surface channels (defined as all linear meltwater features > 1000 m long) to our dataset using the automated methods from Yang et al. (2016). We also slightly adjust the NDWI threshold value in order to delineate shallower slush and better partition lake, channel and slush features.

*In our responses below, we include more detail about channel and slush delineation methods, which we will also add to our revised paper.*

I am also unsure as to the method used for partitioning meltwater that 'drains' and refreezes' as it has been shown in the literature that meltwater can stay active for much longer (i.e., weeks) post surface melt cessation. I think there needs to be some further clarification as to the simplicity of the method and acknowledgement that meltwater can linger for longer post surface melt cessation. Additionally, a figure showing this occurring in imagery (from your manual interpretation) may help you here, which could be placed in your supplementary information.

*We agree that our previous method for partitioning meltwater lake drainage vs. refreeze was overly simplistic and poorly explained. We have now improved this method; please see our extensive response to Reviewer 1 for more details (pages 5-9 of that letter).*

In addition, the link to ice velocity and drainage/refreeze events I believe, at times, is rather subjective.

*We agree. We are careful to use language such as 'appears to perturb ice velocity' within the text. We will ensure that the subjective link to ice velocity is made clearer in the manuscript. Please also see our longer response to Reviewer 1 (pages 4-5 of that letter).*

I provide further comments on these issues below.

**Major comments:**

***Feature extraction and terminology:***

In Figure 1, you provide your mapped supraglacial meltwater features for 2018 and 2019 (only figure presenting these in the main body and supplementary information, which is a shame), however it mostly looks like only SGLs and some slush have been captured, with the caveat of some larger, wider sections of supraglacial rivers (these networks look rather fragmented and incomplete). After searching one of the images used via Copernicus Browser (2019-07-25) and comparing with other studies in this area which have captured supraglacial features, there looks to be a number of rivers and streams not captured, or fragmented, by your method.

This is a shame as you state in your *introduction 'we precisely delineate all surface meltwater features (i.e SGLs > 0.0018 km², as well as rivers and slush)'* (Line 85) and within your methodology you state: *'Our threshold values are lower than those used by Corr et al. (2022), which was a deliberate choice as we wanted to detect shallower meltwater features (including small streams and slush) than those considered in that study'* (Line 150-153). However I do not see many small, shallow streams or continuous river channels captured (perhaps a few within the slush regions at most). It is my understanding that a significant part of this paper was to map *all supraglacial meltwater features*, in particular those of small, shallow size, across the catchment to examine feature distribution, evolution and low-high melt year comparison, with an emphasis on their importance albeit their size. I therefore think there are a number of things that need to be addressed in this manuscript and a few avenues which could be taken to improve the dataset and/or clarity in this paper:

1) As stated already, it looks like you have only captured segments of some of the primary (larger) supraglacial river channels within your drainage network, mostly where they input or output an SGL. Therefore, I think you need to make this clear in your introduction and/or methodology that you only partially-capture these types of channels (i.e., primary rivers) and not smaller stream-type networks (i.e., secondary tributary networks which are shallower and transient) and explain why (method limitation?). I think to aid this, it would help to give brief definitions of these differences as per the literature (e.g., Pitcher and Smith, 2019) in your introduction and then, throughout the

manuscript, make sure you refer to these features as 'rivers', rather than interchangably between 'streams' or 'rivers/streams' to make it clear (as you have in some areas of text).

We apologise for the confusion in our original paper. In summary, 'channels' in our study include both rivers and streams, but we do not differentiate between the two. In our revised paper, we will add the following to the introduction: *'Supraglacial channels can be categorised as rivers or streams (Smith et al., 2015), however in this study we do not differentiate between these two types of channels. Instead, we define all channels as being linear meltwater features that are > 1000 m in length.'*

2) After performing NDWI and subsequent thresholding (via Corr et al., 2022), you state you partition your features into slush and rivers via manual interpretation of geometry and colour. I think an additional figure showing these partitioned features (lakes, slush and rivers) mapped would enable the reader to visually understand and assess their differences (i.e., the distribution of individual meltwater features, the overall collective drainage network characteristics and meltwater drainage behaviour – particularly that of smaller SGLs in lower elevations which are discussed) across the two melt years. This could be completed by including multiple mapped subsets in a new figure (within the results in either *Section 3.1* or *3.2*) showing the distribution and subsequent differences in features across selected dates and/or zoom in sections for both the 2018 and 2019 melt seasons.

We agree that it would be useful to show the meltwater distribution and differences between 2018 and 2019 melt seasons. Below, we have included a revised version of Figure 1, which includes additional zoom in/out sections. This also clearly highlights the detail we are now able to capture by adding additional slush and new channel delineation methods

3) Following on from my previous comment about an additional figure, this would also help show how the network evolves seasonally across the two distinct years. A particular interest, as you discussed, is the drainage of small meltwater features (particularly at lower elevations) and how slush develops in these two years. In terms of slush, so far, I can only see your higher elevation slush in Figure 1 (from your maximum extent). It would be therefore interesting to see how slush develops during the melt season alongside your other features.

We have separated our lake, channel and slush features to produce a revised Figure 3 (below, on page 5). We have also created a new supplementary figure (see page 6 of this letter) that details the seasonal evolution of meltwater features throughout both melt years. We acknowledge that it is difficult to see the detail in this figure, so we will also upload a GIF version of this figure alongside our revised paper.

[Figure]

*REVISED Figure 1: Maximum areal extent of supraglacial meltwater features in (a) 2018 and (b) 2019 within the Russell/Leverett Glacier catchment derived from ArcticDEM (black outline). All meltwater features are superimposed during each melt season in our dataset; slush is light blue, channels are green and lakes are dark blue. Elevation contours from the ArcticDEM are shown in grey (m a.s.l). Background is a true colour Sentinel-2 image acquired on 26/09/2019. Inset depicts the location of the catchment within the southwest GrIS. a(i) depicts a supraglacial channel system, a(ii) shows lakes linked with channels, a(iii) is an example of underdeveloped lakes in the ~1600 m region of the catchment, a(iv) depicts slush and channels in the percolation zone (~1700 m). b(i) shows small lakes close to the margins of the*

*catchment, b(ii) highlights linked channels and lakes, b(iii) shows interconnected lakes, channel and slush, b(iv) depicts high-elevation (~1900 m) slush, channels and the highest elevation lake (1880 m) in our 2019 dataset.*

[Figure]

*REVISED Figure 3: Time series of total areas of lakes (dark blue), channels (green), slush (light blue) and all meltwater features (grey) in (a) 2018 and (b) 2019 from L8 and S2 imagery. Area error bars represent uncertainty of automatically delineated features compared to a manually delineated dataset. Lake volume is given in red error along with an estimate of uncertainty determined by comparing lake depth to Melling et al. (2024). Also shown is cloud cover percentage (black bars), RACMO 2 m air temperature anomaly (light green line) from the 1958 - 2019 catchment average with the spatial standard deviation (light green shading), and RACMO total daily melt (mm w.e.; light blue line) with the spatial standard deviation (light blue shading). Note that the y-axis ranges are different for the channel and slush areas between (a) and (b).*

[Figure]

*NEW Supplementary figure: Areal extent daily snapshots of supraglacial meltwater features in 2018 (a) and 2019 (b) within the Russell/Leverett Glacier catchment derived from ArcticDEM (black outline). Slush is light blue, channels are green, and lakes are dark blue. Elevation contours from the ArcticDEM are shown in grey.(Note, we will also include a GIF version of this figure in the Supplement.)*

4) You could, time- and review-dependent, try to adapt your thresholding method or perform further manual delineation to capture more of the smaller network (i.e., a more complete supraglacial river and stream network) alongside your other smaller features (SGLs) to give a more hollistic view of supraglacial meltwater, and its drainage as a whole, in this catchment. Supraglacial rivers and streams can make up a large portion of the supraglacial network, and so are important to consider alongside your captured, smaller SGLs. This is a suggestion to uphold the use of '*all meltwater features' to* elevate the paper, however, I understand that this may be a considerable undertaking.

We will remove any partially delineated channels from our original data set and replace them with the channels in our new separate channel dataset. To describe the methods we use to now delineate these channels, we will add text along the lines of the following to our revised methods: '*Supraglacial channels have different physical and spectral characteristics to SGLs and thus we delineate channels using methods developed by Yang et al. (2016). We extract channels based on their Gaussian-like cross-sections and longitudinal open-channel morphometry (Lu et al., 2020). Meltwater features are first enhanced by applying an NDWI to the image. A band-pass filter ramped between 1/200 and 1/40 m-1 (Yang et al., 2017) is then applied to remove low frequency background and high frequency noise. This is followed by Gabor filtering (set to < pixels 2 width) to amplify the cross-section of channels. A path opening operator (with a minimum length of 20 pixels) is then implemented for better connectivity. We then remove any features < 1000 m in length to reduce classification uncertainties. A global threshold of 5 (for T22WEV S2 and all L8 tiles) or 10 (for T22WFV S2 tiles) out of 255 was then used to extract the Channels (Rawlins et al., 2023; Lu et al., 2020). Finally, channel features were pologonized and manually cleaned up (e.g. by the removal of previously delineated lake and slush features) before subsequent analysis.*'

Additionally, we have now also added extra detail to the slush dataset by slightly decreasing the NDWI and NDWIice thresholds from 0.25 and 0.24 to 0.15 and 0.14, respectively. This is following Bell et al. (2017) and Yang et al. (2013) who use a lower threshold value to detect slush than we did previously.

**Ice velocity:**

I have no problem with the use of NASA MEaSUREs ITS_LIVE data – it is a well-used and useful data resource. However, there is a lack of acknowledgement of the error and uncertainty regarding this data and implications this may have on your inferred results. For example, a lack of error envelopes for your ice velocity data presented in Figure 5 (particularly important for higher elevation, noisy data). You present an estimate of uncertainty in your Figure 3 – it would be good to do the same in Figure 5.

We have edited Figure 5 (pasted below) to include uncertainty calculations. Please also see our response to Reviewer 1 for more details (particularly pages 4-5 of that letter, which also includes a new supplementary figure).

Secondly, the link between your drainage/refreeze events and impact on ice velocity looks highly subjective and are difficult to intepret and verify from Figure 5 as you do not separate these events by elevation (like you have done with ice velocity) – consider improving this figure.

We have now edited Figure 5 by separating drainage/refreezing events by elevation REVISED Figure 5 below).

[Figure]

*REVISED Figure 5: Time series of lake drainage and refreeze within the Russell/Leverett catchment in a) 2018 and b) 2019. From top to bottom: daily frequency of lake drainage/refreeze events (i.e., the number lakes that drained or refroze); total daily lake area loss; total daily volume loss; mean ice velocity at 800 m a.s.l (red), 1200 m a.s.l (orange), 1600 m a.s.l (blue) and 2000 m a.s.l (purple). Shading indicates the uncertainty taken directly from the ITS_LIVE data product (Gardner et al., 2018; 2023). Vertical grey shaded columns depict, and velocity perturbations discussed in the text (vertical grey shaded columns, labelled i, ii, iii, and iv); Daily values of meltwater discharge through the Watson River (black line) and associated uncertainty (grey shading). Note that the x-axis date ranges are different for each year, constrained by the first and last meltwater feature drainage events in each melt season. Also note that the y-axis for the velocity plots differ between elevation bands.*

You also refer to an ice velocity increase coinciding with a period of refreezing in July 2019 (Line 391). How do these two mechanisms work?

We will remove this statement from the manuscript as we do not have enough information to speculate about these two mechanisms.

**Minor comments:**

*Units:* You interchange your use of units, from km$^2$ to m$^2$. For example, when defining small meltwater features you use km$^2$ (e.g., 0.0495 km$^2$), however when presenting your results, you

refer to your meltwater features in $m^2$. Please choose one unit for consistency and comparison of results.

We will ensure that we have unit consistency in our revised manuscript.

***Catchment reference:*** You refer to the Russell/Leverett catchment throughout the manuscript and provide its outline in Figure 1. Is this catchment delineated yourself (e.g., via flow routing) or is this an already pre-defined catchment? If the former, please provide a method as to how this catchment was delineated and what datasets were used. If the latter, please cite the appropriate data source.

This is a pre-defined catchment created using ArcticDEM, which is stated in our original manuscript, i.e.: '*The surface drainage basin is derived from ArcticDEM Digital Elevation Model at 1 km resolution' (L 100).*' However, we will change 'basin' to 'catchment' to make this clearer.

**Specific comments:**

Line 44 – '*SGLs generally form in early summer enlarge in area and depth between spring and summer as they accumulate water…'.* This sentence does not fully make sense. Maybe add an *'and'* before enlarge or *',enlarging in area…'.*

Done.

Line 60 – could include additional references to remote sensing studies here (Lu et al., 2021; Turton et al., 2021; Rawlins et al., 2023; Zhang et al., 2023).

Done.

Line 73 – '*…the drainage of meltwater features was not considered*.' Drainage how? I am assuming drainage subglacially, but could be more explicit here as some of the papers cited assess how meltwater moves or is 'supraglacially-drained' across the surface over a single/multiple melt seasons.

We will edit this to say: '*the vertical drainage of meltwater from the ice sheet surface to the bed was not considered.*'

Line 95 *(Figure 1)* - Your figure shows maximum areal extent of meltwater features in 2018 and 2019. Is this from a particular date in the season or an amalgamation of your features from separate mapped dates across the season into one map?

We will edit this to say: …'*All meltwater features are superimposed during each melt season in our dataset*'.

Line 102 – This study area is well known for its prevalent surface hydrology features including lakes, rivers, and moulins. Could you additionally provide an upper elevation estimate from previous studies (including refs)?

We will add: '*Supraglacial lakes and channels have been observed to drain into moulins at 1600 m a.s.l in our study area (Yang et al., 2021). Although, it is suggested that drainage by new hydrofracture events is restricted to elevations < 1600 m due to low surface strain rates in higher elevation regions (Poinar et al., 2015)*'.

Line 143 – the blue and green filter you use. What is this? A band combination? Some clarity would help.

We will edit to say: ' …as well as an additional blue and a further green filter.'

Line 146 – Include the citation and subsequent reference for McFeeters (1996) – paper for the traditional NDWI index (using green and NIR bands).

Done.

Line 171 – how narrow were the channels that were manually added? Are they still larger, primary rivers (as commented on earlier)?

See previous comment re: the change in our channel delineation methods.

Line 260 – would be helpful to give a clarifying statement as to the purpose of ice velocity data for the study to make this clearer.

We will : 'We use ITS_LIVE velocity as ancillary data to support our analysis. Its purpose is to infer links between lake drainage events and ice speed up through basal sliding.'

Line 285 – You refer to both linear stream and river features. Maybe just state rivers (as per my previous comments).

As mentioned earlier in this letter (page 7), we are now able to delineate both streams and rivers using our new channel delineation method, however, we do not see the need to differentiate between these two features. As such, we have changed our terminology to 'channels', which we define as river and stream features > 1000 m in length.

Line 412 – Whilst I think it is relatively clear the contrast between surface meltwater characteristics and distribution (which would be helped by an additional figure) between years, I think I would refrain from saying there is a 'clear contrast' for drainage dynamics, as this looks to be subjective.

We will ensure that the subjective nature of the contrast in drainage dynamics between years is represented in the revised manuscript and we will remove any phrasing that suggests otherwise.

Table S1 – From the main body it was stated that 'Images with > 50% cloud cover were omitted' (Line 122). However, in Table S1, scene IDs have been included with cloud cover >50%. Do these scene IDs need to be removed from the table? Or were these scene IDs (>50% cloud) used? If they were used, rectification in the main body is required.

We initially limited the cloud cover to < 50% based on the image metadata. However, after manual inspection, it appeared that the cloud cover algorithm used to create the S2/L8 image metadata misclassified some white ice/snow as cloud. Therefore, we manually checked all available images and included misclassified images in our dataset. We will update the text in our manuscript to explain this.

**Technical corrections:**

Line 126 – Capitalise 'Level' Done.

Line 131 – Missing bracket for citation Added.

Line 133 – decide whether to capitalise (or not) the word 'bands'. Some inconsistencies. Fixed.

Line 139 – you have already given the abbreviation for Normalised Difference Water Index (NDWI) on Line 89. You could therefore remove this on Line 139 if you wish (however, if retaining, please capitalise 'Normalised Difference Water Index' for consistency). Fixed.

Line 217 – replace 'ocean' with lake or SGL

'Ocean' is correct. See: Sneed and Hamilton (2007). We have added this reference to the text.

Line 296 – At the end of the sentence, either remove 'at' or the brackets.

We do not see this issue on line 296, so we wonder if the reviewer has the line number wrong.

In places, you have a citation and Figure number in separate brackets in immediate succession. Combine these using a semi-colon (Corr et al., 2022; Fig. 2).

Done.

References not in the original paper (which will be added to our revised paper)

Bell, R., Chu, W., Kingslake, J. *et al.* Antarctic ice shelf potentially stabilized by export of meltwater in surface river. *Nature* 544, 344–348 (2017). https://doi.org/10.1038/nature22048

Lu, Y., Yang, K., Lu, X., Li, Y., Gao, S., Mao, W. and Li, M.: Response of supraglacial rivers and lakes to ice flow and surface melt on the Northeast Greenland ice sheet during the 2017 melt season, J. Hydrol. 602, 126750, 2021. https://doi.org/10.1016/j.jhydrol.2021.126750

McFEETERS, S. K. (1996) 'The use of the Normalized Difference Water Index (NDWI) in the delineation of open water features', International Journal of Remote Sensing, 17(7), pp. 1425–1432. https://doi.org/10.1080/01431169608948714.

Sneed, W. and Hamilton, G.: Validation of a method for determining the depth of glacial melt ponds using satellite imagery, Ann. Glaciol., 52, 15–22, https://doi.org/10.3189/172756411799096240, 2011

Turton, J. V., Hochreuther, P., Reimann, N., and Blau, M. T.: The distribution and evolution of supraglacial lakes on 79° N Glacier (north-eastern Greenland) and interannual climatic controls, The Cryosphere, 15, 3877–3896, https://doi.org/10.5194/tc-15-3877-2021, 2021. https://doi.org/10.5194/tc-15-3877-2021